# Trajectory-level Data Generation with Better Alignment for Offline Imitation Learning

## Abstract

Offline reinforcement learning (RL) relies heavily on densely precise reward signals, which are labor-intensive and challenging to obtain in many real-world scenarios. To tackle this challenge, offline imitation learning (IL) extracts optimal policies from expert demonstrations and datasets without reward labels. However, the scarcity of expert data and the abundance of suboptimal trajectories within the dataset impede the application of supervised learning methods like behavior cloning (BC). While previous research has focused on learning importance weights for BC or reward functions to integrate with offline RL algorithms, these approaches often result in suboptimal policy performance due to training instabilities and inaccuracies in learned weights or rewards. To address this problem, we introduce Trajectory-level Data Generation with Better Alignment (TDGBA), an algorithm that leverages alignment measures between unlabeled trajectories and expert demonstrations to guide a diffusion model in generating highly aligned trajectories. The aforementioned trajectories allow for the direct application of BC in order to extract optimal policies, negating the necessity for weight or reward learning. In particular, we define implicit expert preferences and, without the necessity of an additional human preference dataset, effectively utilise expert demonstrations to identify the preferences of unlabeled trajectories. Experimental results on the D4RL benchmarks demonstrate that TDGBA significantly outperforms state-of-the-art BC-based IL methods. Furthermore, we illustrate the efficacy of implicit expert preferences, which represents the inaugural application of the benefits of preference learning to offline IL.

## 1 Introduction

Reinforcement Learning (RL) has achieved remarkable success in a diverse range of domains, including video games (Silver et al., 2017), large-scale language modeling (Ouyang et al., 2022), protein structure prediction (Lutz et al., 2023), among others. While current RL techniques typically require dynamic interactions with an environment and immediate reward feedback for policy learning, such online interactions can be costly, time-consuming, and risky in many practical scenarios. To mitigate these challenges, offline RL (Lutz et al., 2023) leverages pre-collected datasets with densely annotated rewards to derive optimal policies (Kostrikov et al., 2021; Fujimoto & Gu, 2021; Kumar et al., 2020). Although offline RL has led to significant advancements in various applications, the design of accurate and robust reward functions remains a complex, laborious task, posing substantial difficulties in real-world settings.

To address this issue, with a small set of expert demonstrations and the offline datasets where no reward signals are provided, offline imitation learning (IL) (Rashidinejad et al., 2021) has been proposed to extract policies by utilizing behavior cloning (BC) (Pomerleau, 1988) or inverse reinforcement learning(IRL) (Jarboui & Perchet, 2021). However, their performance is closely related to the quality of the datasets and the expert demonstrations. For the BC-based approaches, the prevalence of suboptimal trajectories in offline datasets may hinder their ability to extract optimal policies due to the distributional shift problem (Ross et al., 2011). To alleviate this problem, distribution divergence measures (e.g., $f$-divergence (Ghasemipour et al., 2020)) are employed to align learned policies with behavioral policies and a discriminator is trained to determine the optimal importance weights for weighted BC (Ma et al., 2022; Kim et al., 2022; 2021). However, such distribution matching schemes can incur training instability (Pomerleau, 1988) and over-conservatism (Lyu et al., 2024). Some work

utilizes expert demonstrations to learn intermediate reward functions (Lyu et al., 2024; Zolna et al., 2020; Luo et al., 2023), which are then incorporated with existing offline RL methods. However, offline RL is inherently challenging due to its sensitivity to hyperparameters, and learning accurate transition-wise rewards is also difficult.

It has been demonstrated that when training data includes sufficient expert knowledge, learning an effective policy becomes relatively easy (Xu et al., 2022). While some studies have explored expanding expert data through data augmentation, these methods often face challenges such as overfitting, stemming from the limited availability of expert data and the insufficient diversity and fidelity of generated samples (Sun et al., 2023). Furthermore, a significant amount of unlabeled trajectory data often remains underutilized. Recently, Flow-to-Better (FTB) (Zhang et al., 2023) introduced a method that combines human preferences across diverse trajectories with a trajectory diffuser to iteratively refine and enhance trajectories, improving both the quality and diversity of training data. However, collecting task-specific human preference data is inherently time-consuming and resource-intensive. This raises an intriguing question: *Could we construct an expert-like preference from limited expert data to guide trajectory generation efficiently?*

Based on the above introduction, we propose a method named Trajectory-level Data Generation with Better Alignment (TDGBA). Firstly, we propose a simple and efficient alignment measurement based on optimal transport theory and Wasserstein distance. Trajectories with small distances are referred to as high-alignment trajectories, and those with large distances are referred to as low-alignment trajectories, with the alignment measurement defined as *implicit expert preference*. Then, we adopt the trajectory optimization approach proposed in FTB, which trains a trajectory diffuser to refine low-alignment trajectories, gradually aligning them with expert trajectories. The generated diverse and near-expert trajectories are then used to enable the agent to learn policies through a straightforward behavior cloning (BC) method. The main contributions are as follows:

1) We present a novel framework for offline IL that can learn optimal policies directly from generated trajectories without the need of learning reward functions or behavior weights;

2) By introducing the implicit expert preference, high-fidelity and diverse trajectories can be generated with a trajectory diffuser, even with only a single expert trajectory;

3) Experimental results show that TDGBA outperforms previous offline IL methods across multiple tasks. Additionally, the effectiveness of our method provides the first evidence of the potential and feasibility of applying the advantages of preference learning to the offline IL domain.

## 2 RELATED WORK

Offline Imitation learning learns policies from few expert demonstrations without real rewards. By using supervised learning, Behavioral Cloning (BC)(Pomerleau, 1988) is one of the simplest IL methods. However, in datasets with insufficient expert demonstrations, challenges such as compounded errors and inadequate generalization capabilities often arise. To alleviate these problems, BCND (Sasaki & Yamashina, 2020) reuses other BC policies as target weights for the original BC policy. Recently, DICE (distribution divergence measure)-related methods (Ma et al., 2022; Kim et al., 2022; 2021) have garnered attention by transforming the min-max problem in adversarial IL into a convex optimization problem. However, these methods usually suffer from training instability and excessive conservatism. Another line of the offline IL research is offline inverse RL(IRL) (Jarboui & Perchet, 2021), which learns reward functions from expert demonstrations and then existing offline RL algorithms can be used for the policy training. Some typical work of IRL includes ORIL (Zolna et al., 2020), CLARE (Yue et al., 2023) and IQ-learn (Garg et al., 2021), where ORIL (Zolna et al., 2020) applies the online inverse RL method GAIL (Ho & Ermon, 2016) to the offline setting, CLARE introduces the maximum entropy inverse reinforcement learning framework to the offline setting, and IQ-learn (Garg et al., 2021)implicitly learns rewards and policies by learning the Soft-Q function. By decoupling reward learning and policy optimization, the optimal transport distance or the nearest neighbor distance is used in OTR (Luo et al., 2023) and SEABO (Lyu et al., 2024) as rewards. However, the reward distribution learned by these methods often significantly differs from the actual reward distribution, making policy learning susceptible to inaccuracies in rewards and sensitivity to hyperparameters. In recent years, several offline preference reinforcement learning methods have alleviated the challenge of reward design by introducing preference learning. Methods

such as ORPL(Shin et al., 2023) and PT(Kim et al., 2023) assign rewards to transitions by learning a score model. Additionally, FTB(Zhang et al., 2023) proposes a trajectory optimization scheme based on the trajectory diffuser and trajectory preferences, achieving state-of-the-art performance in many tasks. However, these methods rely on the availability of an additional, carefully designed human preference dataset, which imposes a critical limitation. Due to this requirement, the advantages of preference learning cannot be extended to offline imitation learning scenarios where only a small number of expert demonstrations are available. To address this issue, we propose the key concept of implicit expert preferences and a method for measuring them. By integrating these with the trajectory optimization approach in FTB(Zhang et al., 2023), we achieve significant breakthroughs in performance of offline IL. To the best of our knowledge, we are the first to successfully transfer the advantages of preference learning to offline imitation learning. We believe that TDGBA will offer researchers a promising new direction for exploration.

## 3 PRELIMINARIES

**Offline Imitation Learning**  We consider the interaction between the environment and policy as a Markov Decision Process (MDP) (Puterman, 2014), defined by a tuple $(\mathcal{S}, \mathcal{A}, \mathcal{P}, \rho, \mathcal{R}, \gamma)$ representing states, actions, the transition probability of the environment, the initial state distribution, the reward function, and the discount factor. Reinforcement learning (RL) learns a policy $\pi_\theta(a|s)$ to maximize the discounted cumulative reward $J(\theta) = \mathbb{E}_{\pi_\theta}[\sum_{t=0}^{T-1} \gamma^t \mathcal{R}(s_t, a_t)]$. In offline IL setting, the ground truth reward function is not observed. Instead, we assume that we have access to a set of expert demonstrations $\mathcal{D}^E = \{\tau_m^e\}_{m=1}^M$ and a dataset of unlabeled trajectories $\mathcal{D}^U = \{\tau_n^u\}_{n=1}^N$, where $M$ and $N$ are the sizes of trajectories in the expert dataset and unlabeled dataset. Note that unlabeled trajectories are collected from some sub-optimal behavior policies $\pi_\beta$. By leveraging the offline dataset $\mathcal{D}^E \cup \mathcal{D}^U$, we learn the optimal policy without any interaction.

**Wasserstein distance**  The Wasserstein distance is a method for measuring the difference between distributions which captures the geometry of the underlying space and does not require any intersection between the support sets(Panaretos & Zemel, 2019; Luo et al., 2023). Let $\Gamma(p, q)$ denotes the set of all distributions in the product space $X \times Y$, where the marginal distributions of $X$ and $Y$ are $p$ and $q$ respectively. Given an appropriate cost function for representing the cost of moving a unit mass from $x$ to $y$. Thus, the original format of the Wasserstein distance can be defined as: $W(p, q) = \inf_{\gamma \in \Gamma_{[p,q]}} \int_{X \times Y} c(x, y) \, d\gamma$, where the family of joint distributions $\Gamma(p, q)$ essentially forms a series of bijective plans for transferring the probability distributions $p$ to probability distributions $q$. For two discrete probability distributions $\mu_x = \frac{1}{T} \sum_{t=1}^{T} \delta_{x_t}$ and $\mu_y = \frac{1}{T'} \sum_{t'=1}^{T'} \delta_{y_{t'}}$, the Wasserstein distance calculation is changed to: $W(p, q) = \min_{\mu \in M} \sum_{t=1}^{T} \sum_{t'=1}^{T'} c(x_t, y_{t'}) \mu_{t,t'}$, where $M = \{\mu \in \mathbb{R}^{T \times T'} : \mu \mathbf{1} = \frac{1}{T} \mathbf{1}, \mu^T \mathbf{1} = \frac{1}{T'} \mathbf{1}\}$ denotes the set of coupling matrices, $\delta_x$ and $\delta_y$ are the Dirac measures respectively. The optimal coupling matrix $\mu^*$ represents the best alignment between these two discrete distributions.

**Flow-to-Better with Trajectory Diffuser**  Flow-to-Better (FTB) (Zhang et al., 2023) is a state-of-the-art approach in offline preference reinforcement learning, introducing a trajectory diffuser to progressively refine low-preference trajectories into high-preference ones. In this field, a dataset with human-preference labels, denoted as $\mathcal{D}^{human} = \{(\tau_m^0, \tau_m^1, y_m)\}_{m=1}^M$, needs to be pre-collected. $y_m \in \{0, 0.5, 1\}$ serves as the preference label, where 0 implies that $\tau_m^0 \succ \tau_m^1$, 1 implies $\tau_m^1 \succ \tau_m^0$, and 0.5 implies $\tau_m^0 \sim \tau_m^1$. Leveraging this prior dataset, a preference model $s_\psi$ is trained following the Bradley-Terry model (Bradley & Terry, 1952):$P[\tau_m^1 \succ \tau_m^0] = \frac{exp \, s_\psi(\tau^1)}{exp \, s_\psi(\tau^0) + exp \, s_\psi(\tau^1)}$. After training, the preference scores of unlabeled trajectories are computed:$pref(\tau^u) = s_\psi(\tau^u)$. Labeled trajectories are clustered into $K$ blocks based on their preferences, with the blocks arranged in ascending order $B_1 \prec \cdots \prec B_K$. FTB formulates the trajectory optimization problem as a conditional generation task, treating lower-preference trajectories as the condition. The objective is to maximize the conditional log-likelihood of generating optimized higher-preference trajectories:

$$\min_\theta \mathbb{E}_{(\tau^1 \succeq \tau^0) \sim \mathcal{D}^U}[-\log p_\theta(\tau^1 | \tau^0)] \tag{1}$$

where $\tau^0$ is the low-preference conditional trajectory, and $\tau^1$ is the high-preference target trajectory. To address the multimodal target distribution issue among trajectory pairs with varying preferences,

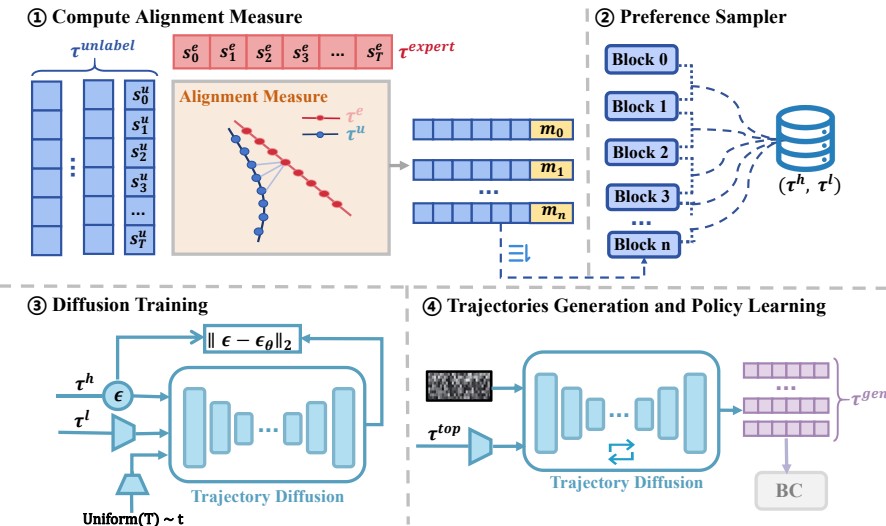

Figure 1: Overview of TDGBA. We begin by calculating alignment measures and then extract implicit expert preferences. Subsequently, input low-alignment trajectories for guiding the diffusion model $\epsilon_\theta$ to generate high-alignment trajectories. Finally, high-quality trajectories are generated for BC.

the authors proposed a classifier-free diffusion model(Nichol et al., 2021), *Trajectory Diffuser*, as the generative model. They further incorporated an neighboring sampling strategy to formulate the final training loss for the model:

$$\min_\theta \frac{1}{K-1} \sum_{k=1}^{K-1} \mathbb{E}_{\substack{t,\epsilon,\beta\sim\text{Bern}(p) \\ \tau^0\sim B_k, \tau^1\sim B_{k+1}}} \left[ \left\| \epsilon - \epsilon_\theta(\tau_t^1, (1-\beta)\tau^0 + \beta\varnothing, t) \right\|^2 \right] \qquad (2)$$

where $t$ is the diffusion timestep sampled from $t \sim \mathcal{U}\{1,\ldots,T\}$, $\tau_t^1$ is the high-preference target trajectory in the t diffusion timestep, $\epsilon_\theta$ is used to predict the perturbation noise added from $\tau_{t-1}^1$ to $\tau_t^1$, $\epsilon$ is sampled from the Gaussian distribution $\mathcal{N}(0,\mathcal{I})$. Furthermore, $\beta \sim \text{Bern}(p)$ indicates the probability $p$ used to mask the condition, and $\varnothing$ serves as a placeholder for the condition $\tau^0$.

## 4 METHOD

In this section, we present the Trajectory-level Data Generation with Better Alignment (TDGBA) approach for offline IL, which can measure the alignment between a trajectory and expert demonstrations and generate high-quality trajectory-level training data with diffusion process for policy learning. The overall process of our method is depicted in Figure 1. First, we show how to compute the alignment measurement between an unlabeled trajectory and expert demonstrations 4.1. Next, we define implicit expert preferences and examine their feasibility and advantages in trajectory labeling (4.2). The labeled trajectories are subsequently fed into the trajectory optimization process in FTB(Zhang et al., 2023), enabling the generation of high-quality trajectories for learning the optimal policy 4.3.

### 4.1 COMPUTE ALIGNMENT MEASUREMENT

For an unlabeled trajectory in the dataset, it is clear that its alignment measurement with expert demonstrations determines the quality in BC-based policy learning. However, to our best knowledge, how to measure the alignment in a trajectory level is still unsolved in offline IL. Recent works have computed the Wasserstein distance between trajectories and assigned rewards to each transition, which are then used to learn policies via reinforcement learning methods, resulting in some performance improvement (Luo et al., 2023)(Dadashi et al., 2020). However, this reward assignment approach has

been found to suffer from issues such as large discrepancies with the true reward distribution and incorrect reward labeling (Fu et al., 2024)(Liu et al., 2024). Specifically, in the experimental section, we visualize the reward distribution differences of such methods and provide additional performance comparisons. We hypothesize that the root cause of these issues lies in the fact that Wasserstein distance inherently measures the discrepancy between distributions, and assigning this discrepancy as rewards to each transition is a challenging and error-prone task.

In our approach, the Wasserstein distance is utilized for computing the alignment measurement between unlabeled trajectories and expert demonstrations. In details, the Wasserstein distance between two trajectories is computed as below:

$$W(\tau_u, \tau_e) = \min_{\mu} \sum_{i=1}^{T} \sum_{j=1}^{T} c(s_i^u, s_j^e)\mu_{i,j} \tag{3}$$

With the solution of Eq. 3, which is treated as an optimal transport problem, the optimal coupling matrix $\mu_{i,j}^*$ can be obtained and the alignment measurement between the unlabeled trajectory $\tau^u$ and expert demonstration $\tau^e$ can be calculated. Furthermore, considering the different data characteristics of various tasks and drawing inspiration from (Cai et al., 2019)(Luo et al., 2023), we introduce a scaling function to maintain the alignment measure within an appropriate range and ensure its smoothness. Finally, we update the calculation equation for the alignment measurement as follows:

$$m(\tau^u, \tau^e) = \sum_{i=1}^{T} \sum_{j=1}^{T} f(c(s_i^u, s_j^e)\mu_{i,j}^*) \tag{4}$$

where $f$ in Eq. 4 denotes the scaling function (e.g,$f(x) = exp(\beta x)$). Once the alignment measurements of all trajectories are computed, they are directly used to label the dataset $\mathcal{D}^U = \{(\tau_n, m_n)\}_{n=1}^{N}$ without reward labeling.

### 4.2 IMPLICIT EXPERT PREFERENCES

The alignment measurement mentioned in the previous section reflects the distance between different trajectories and expert demonstrations. In a sense, this distance represents the degree of preference human experts have for a given trajectory, which we define as *implicit expert inference*.

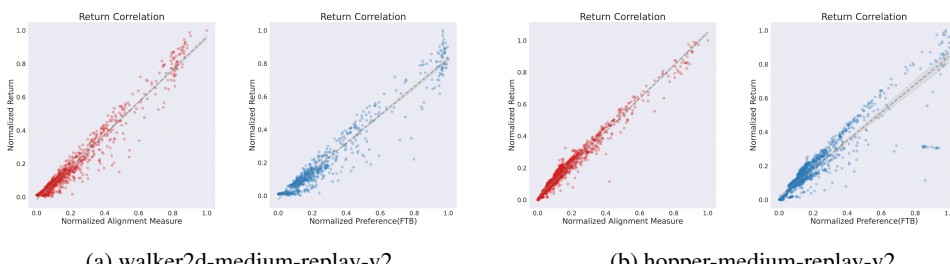

(a) walker2d-medium-replay-v2          (b) hopper-medium-replay-v2

Figure 2: Comparison of the Correlation. We illustrate the correlation between the ground-truth returns of each trajectory in the offline dataset and two metrics across two tasks: the alignment measure (red scatter points) and the preferences labeled by the preference model trained with FTB (blue scatter points).

To validate whether the alignment measure can be used as an implicit expert preference for labeling unlabeled trajectories, we selected two locomotion tasks (hopper, walker2d) and visualized the correlation between this metric and the ground-truth trajectory returns in Figure2. Additionally, we further presented the correlation between the preference calculated by the preference model trained using a prior human preference set in FTB and the ground-truth returns. Notably, we selected only one expert trajectory for expert demonstration, while in FTB, we followed the original paper's setting and used 15 trajectory pairs for training. The results show that our metric has a strong correlation with the ground-truth returns, and even with just one expert trajectory, it achieves or surpasses the performance of training with multiple human preferences. Additionally, we evaluated the accuracy

Table 1: Accuracy of trajectory evaluation by the preference model trained with different numbers of prior human preference trajectories.

| Task Name | num=15 | num=50 | num=100 | num=150 | num=200 |
|---|---|---|---|---|---|
| hopper-medium-replay-v2 | 0.93 | 0.82 | 0.83 | 0.86 | 0.87 |
| walker2d-medium-replay-v2 | 0.79 | 0.76 | 0.87 | 0.92 | 0.89 |

of the preference model by re-scoring the trajectory pairs from the prior human preference dataset using the trained model and labeling the preference values in $\{0, 0.5, 1\}$. We then compared these scores with the original ground-truth labels to calculate the accuracy. Furthermore, we compared the model's accuracy across different numbers of human preference trajectories. As shown in Table 1, the accuracy of the preference model fluctuates with the number of prior preference trajectory pairs, and in some cases, it even decreases. We hypothesize that as the number of preference pairs increases, the diversity of preference types may also rise, potentially introducing detrimental noise, thereby complicating the model training process.

### 4.3 TRAJECTORY OPTIMIZATION WITH TRAJECTOR DIFFUSER AND POLICY LEARNING

Based on the above analysis, we found that implicit expert preferences can effectively estimate the preference of different trajectories. Therefore, we further integrate this preference with the trajectory optimization and behavior cloning methods proposed in FTB. Algorithm 1 gives a formal description of TDGBA.

---

**Algorithm 1** Trajectory-level Data Generation with Better Alignment (TDGBA)

---

**Require:** expert demonstrations $\mathcal{D}^E = \{\tau_m^e\}_{m=1}^M$, unlabeled dataset $\mathcal{D}^U = \{\tau_n^u\}_{n=1}^N$,
  number of blocks $K$, number of top trajectories $\mathcal{Z}$, blocks $B = \{B_1, \cdots, B_K\}$
 1: # Compute implicit expert preference.
 2: Initialize policy net $\pi_\phi$, generative model $p_\theta$ .
 3: **for** $\tau_n^u$ in $\mathcal{D}^U$ **do**
 4:     $m_n \leftarrow \text{AlignmentMeasure}(\tau_n^u, \tau^e)$
 5:     label preferences $\mathcal{D}^U \leftarrow \{(\tau_n^u, m_n)\}$
 6: **end for**
 7: # Train a Trajectory Diffuser and generate trajectories like FTB.
 8: Cluster $\mathcal{D}^U$ based on preferences: $\mathcal{D}^U = B_1 \cup B_2 \cup \ldots \cup B_K$ where $B_1 \prec B_2 \prec \ldots \prec B_K$.
 9: Collect preference trajectory pairs: $\{(\tau^0, \tau^1)\} \sim \text{Neighbor-Block-Sample}(\mathcal{D}^U, B)$.
10: Iteratively update the generative model $p_\theta$ using batched trajectory pairs.
11: Select the top $\mathcal{Z}$ trajectories based on preferences.
12: Feed top trajectories into model $p_\theta$ to optimize and generate new trajectories.
13: # Trajectory optimization is completed for policy learning.
14: Behavior Cloning with generated trajectories.

---

## 5 EXPERIMENT

In this section, experiments are conducted to evaluate the performances of TDGBA in different tasks and we demonstrate the advantages of it over SOTA offline IL methods including both the BC-based approaches and the reward-based approaches that combine offline RL and the learned rewards. Experimental results show that TDGBA outperforms these algorithms in most cases, verifying the benefits of the preference-induced trajectory-level data generation. Then, we confirm that higher quality and more diverse samples can be obtained with the utilization of implicit expert preferences for the diffusion model training. Additionally, we conducted supplementary experiments to compare different parameter settings.

### 5.1 COMPARISON WITH BC-BASED IMITATION LEARNING ALGORITHMS

**Setup** We evaluate the performance of TDGBA on the D4RL benchmark (Fu et al., 2020) (HalfCheetah-v2, Walker-v2, Hopper-v2). We discard reward signals in the D4RL datasets to

form unlabeled datasets. For the expert demonstrations, we follow the settings in the previous work (Lyu et al., 2024; Luo et al., 2023) and utilize the trajectory with the highest return in the raw dataset for fair evaluation. For computing the Wasserstein Distance, we follow the recommendation by (Luo et al., 2023; Cai et al., 2019) and use cosine distance as the cost function. In addition, we utilize the Sinkhorn algorithm from the optimal transport library OTT-JAX (Cuturi et al., 2022) to compute the optimal coupling matrix, thereby enhancing computational efficiency. We refer to Appendix A.1 for additional experimental details and hyperparameters.

**Baselines** We compare algorithms including DemoDICE (Kim et al., 2021), SMODICE (Ma et al., 2022), and LobsDICE (Kim et al., 2022), the three most powerful algorithms in the DICE family. In addition, we convert the online IL algorithm SQIL (Reddy et al., 2019) to an offline setup while replacing the algorithm in SQIL with TD3BC. Furthermore, we compare BC and %10BC (Chen et al., 2021) with ground-truth returns.

Table 2: D4RL performance comparison against BC-related algorithms with a single expert demonstration. We report the mean normalized score averaged over 5 random seeds and standard deviation, then bold the highest score for each task.

| Task Name | BC | %10BC | SQIL | DEMODICE | SMODICE | LobsDICE | TDGBA |
|---|---|---|---|---|---|---|---|
| halfcheetah-medium-v2 | 42.6 | 42.5 | 31.3±6.2 | 42.5±1.7 | 41.7±3.9 | 41.5±0.9 | **42.8±1.4** |
| hopper-medium-v2 | 52.9 | 56.9 | 44.7±5.6 | 55.1±2.9 | 56.3±4.0 | 56.9±3.2 | **58.1±8.2** |
| walker2d-medium-v2 | 75.3 | 75.0 | 59.6±7.7 | 73.4±4.9 | 13.3±6.8 | 69.3±2.7 | **76.5±4.6** |
| halfcheetah-medium-replay-v2 | 36.6 | 40.6 | 29.3±5.1 | 38.1±0.3 | 38.7±1.9 | 39.9±2.2 | **42.7±0.5** |
| hopper-medium-replay-v2 | 18.1 | 75.9 | 45.2±5.3 | 39.0±15.4 | 44.3±9.7 | 41.6±10.2 | **93.3±3.9** |
| walker2d-medium-replay-v2 | 26.0 | 62.5 | 36.3±4.9 | 52.2±12.2 | 44.6±2.5 | 33.2±8.9 | **72.8±3.7** |
| halfcheetah-medium-expert-v2 | 55.2 | 92.9 | 40.1±4.7 | 85.8±5.7 | 87.9±3.5 | 89.4±2.1 | **90.1±1.5** |
| hopper-medium-expert-v2 | 52.5 | 110.9 | 49.4±1.5 | 76.0±10.1 | 53.9±6.7 | 44.7±6.2 | **108.9±2.8** |
| walker2d-medium-expert-v2 | 107.5 | 109.0 | 35.9±2.8 | 106.9±1.9 | 47.8±4.3 | 106.6±0.6 | **109.8±0.4** |
| Average Score | 51.9 | 74.0 | 41.4 | 65.0 | 49.0 | 59.1 | **77.2** |

**Results** We conduct experiments on 9 medium-level (medium, medium-replay, medium-expert) D4RL Mujoco locomotion "-v2" datasets and summarize the comparison results in Table 2. It can be found that TDGBA achieves the best performance in all 9 tasks, and the average score is 18.74% ahead of the second highest score. These indicate that TDGBA can perform effectively using direct BC without the need to learn importance weights.

Table 3: Performance comparison of TDGBA against reward leaning methods averaged over 5 random seeds with one single expert demonstration. We report the mean score at the final evaluation for each algorithm, ± captures the standard deviation.

| Task Name | IQL | ORIL | UDS | OTR | SEABO | TDGBA |
|---|---|---|---|---|---|---|
| halfcheetah-medium-v2 | 47.4 | **49.0±0.2** | 42.4±0.3 | 43.2±0.2 | 44.9±0.2 | 42.8±1.4 |
| hopper-medium-v2 | 66.2 | 47.0±4.0 | 54.5±3.0 | 74.9±3.9 | **75.0±5.0** | 58.1±8.2 |
| walker2d-medium-v2 | 78.3 | 61.9±6.6 | 68.9±6.2 | 79.0±1.1 | **79.8±1.5** | 76.5±4.6 |
| halfcheetah-medium-replay-v2 | 44.2 | **44.1±0.6** | 37.9±2.4 | 41.6±0.5 | 42.8±0.8 | **42.7±0.5** |
| hopper-medium-replay-v2 | 94.7 | 82.4±1.7 | 49.3±22.7 | 84.8±1.1 | 86.7±0.3 | **93.3±3.9** |
| walker2d-medium-replay-v2 | 73.8 | **76.3±4.9** | 17.7±9.6 | 64.5±5.9 | 74.9±2.2 | 72.8±3.7 |
| halfcheetah-medium-expert-v2 | 86.7 | 87.5±3.9 | 63.0±5.7 | 87.6±0.8 | **91.8±3.0** | 90.1±1.5 |
| hopper-medium-expert-v2 | 91.5 | 29.7±22.2 | 53.9±2.5 | 81.1±2.5 | 108.9±0.8 | **108.9±2.8** |
| walker2d-medium-expert-v2 | 109.6 | 110.6±0.6 | 107.5±1.7 | 109.6±0.2 | **110.8±0.7** | 109.8±0.4 |
| antmaze-umaze-v0 | 87.5 | 16.7±10.9 | 85.3±2.1 | 84.0±2.3 | 85.6±5.8 | **90.6±4.2** |
| antmaze-umaze-diverse-v0 | 62.2 | 11.4±5.5 | 62.9±3.0 | 71.8±2.5 | 53.4±12.7 | **72.3±6.5** |
| Average Score | 76.6 | 56.1 | 58.4 | 74.7 | 76.9 | **78.0** |

## 5.2 COMPARISON WITH REWARD-BASED IMITATION LEARNING ALGORITHMS

In addition, recent approaches that integrate reward learning with offline RL have shown promising performance in certain tasks. In this subsection, we use D4RL locomotion tasks and the more challenging antmaze tasks. Our proposed approach not only achieves better final performance but

also demonstrates its effectiveness in these tasks. We also find that for reward-based offline imitation learning methods, the learned reward function distribution is significantly shifted from the underlying ground-truth reward distribution.

**Baselines**  We select the following baselines: 1) ORIL (Zolna et al., 2020), an inverse reinforcement learning algorithm that learns a discriminator to output the rewards; 2)UDS (Yu et al., 2022) keeps the ground-truth rewards in the expert demonstrations and simply assigns the minimum rewards to unlabeled data; 3) OTR (Luo et al., 2023) calculates the optimal transfer distance as rewards; 4) SEABO (Lyu et al., 2024) sets the reward function based on KD-tree search, which is a state-of-the art algorithm among the recent offline IL methods. For the above algorithms, we uniformly choose IQL (Kostrikov et al., 2021) as their base offline RL algorithm. Besides, we also compare the IQL algorithm with ground-truth rewards.

**Results**  We present the comparison results of TDGBA against reward learning and offline IL algorithms in Table 3, which show that our method leads or closely matches the best performance in 7 out of 11 tasks and surpasses other algorithms in terms of overall average score. Note that as in the previous work, BC-based imitation learning methods have obviously lower performances when compared with reward-based approach, but as shown in the results, even with no rewards provided, our BC-based algorithm can still achieve a competitive performance, which in some extent means a significant progress in the BC-based offline IL research.

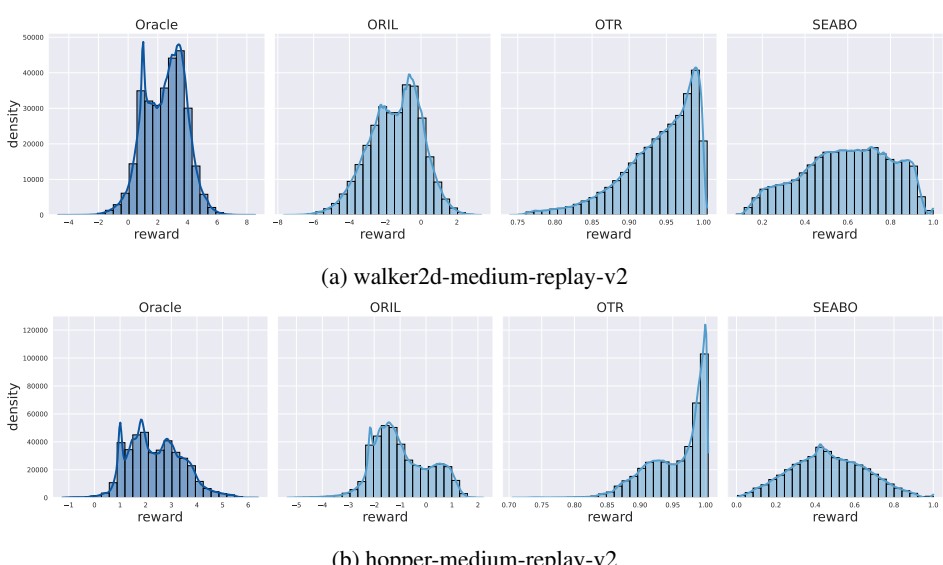

(a) walker2d-medium-replay-v2

(b) hopper-medium-replay-v2

Figure 3: Density plots of ground-truth rewards and learned rewards.

**Analysis**  We argue that reward-based learning is difficult to fit the real reward distribution and often suffer from inaccuracies. In contrast, as shown in Figure 2, we can effectively utilize alignment measures that more closely correspond with ground-truth trajectory returns. To substantiate this claim, we compare the learned rewards form three strong methods with the ground-truth rewards. It can be found that none of these methods can fully fit the original reward distribution in Figure 3. We provided supplementary visualization results for more tasks in Figures 5 and 6 in the appendix.

## 5.3 GENERATED TRAJECTORIES ANALYSIS

In this section, we refer to and further optimize the experimental design from FTB(Zhang et al., 2023) to validate the diversity, accuracy, and effectiveness of the trajectories generated by TDGBA. We also select 300 top-ranked trajectories from the dataset based on their alignment measures and use these trajectories to generate high-alignment trajectories. Additionally, to demonstrate the effectiveness of preference information, we train an unconditioned diffusion model to directly fit the distribution of these trajectories and generate new samples. We compare the groun-truth returns and the performance

of policies trained on these three types of trajectories. As illustrated in Figure 4a, it illustrates the improvement of TDGBA in training generative models with implicit expert preference information.

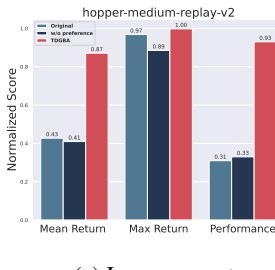
(a) Improvement

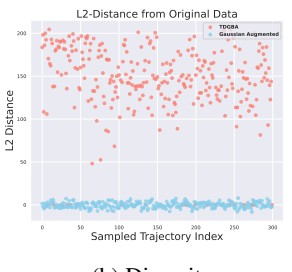
(b) Diversity

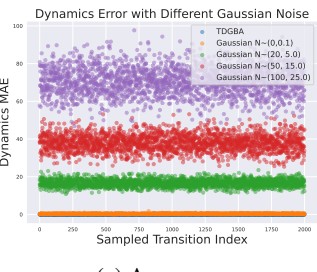
(c) Accuracy

Figure 4: Illustration comparing the average, maximum returns, and final performance of TDGBA-generated versus original trajectories. Additionally, ablation experiments for implicit expert preferences are included (a). Comparison using Gaussian data augmentation to validate the diversity (b) and dynamic accuracy (c) of the generated trajectories.

Moreover, we compare the L2 distances between the generated trajectories and raw trajectories as $L_2(\tau^{gen}) = \min_{\tau \in D^U} \parallel \tau^{gen} - \tau \parallel_2$. We also compare with Gaussian augmented $\mathcal{N}(0, 0.1)$ refer to (Lu et al., 2024). As shown in Figure 4b, the mean and variance of the L2 distance distribution for the generated trajectories are significantly higher than those of the Gaussian perturbation samples. In addition, another key factor in evaluating the quality of generated trajectories is dynamic accuracy. Thus, in Figure 4c, we compare the dynamic error, measured as the mean absolute error (MAE), of trajectories generated by different Gaussian augmentation schemes with varying means and variances against those generated by TDGBA. We observe that as the Gaussian noise increases, the dynamic error rises significantly. In contrast, TDGBA, while maintaining diversity, exhibits lower dynamic error compared to even minimal Gaussian noise augmentation.

## 5.4 COMPARISON TO PWIL AND OTR

In this section, we especially compare two offline IL methods, OTR (Luo et al., 2023) and PWIL (Dadashi et al., 2020), that learn intermediate reward functions using Wasserstein Distance. Unlike them, we maintain this metric only at the trajectory level, avoiding the learning of transition-wise rewards. We ran OTR and PWIL using only states (denoted as OTR-state and PWIL-state) and OTR and PWIL using state-action pairs (denoted as OTR-action and PWIL-action). The results are illustrated in Table 4. We found that TDGBA outperforms both methods across a variety of tasks, further demonstrating the advantage of using Wasserstein Distance for aligning trajectories.

Table 4: D4RL performance comparison to TDGBA, OTR and PWIL. We report mean $\pm$ standard deviation per task and aggregate performance and highlight optimal performance in **bold**.

| Task Name | PWIL-action | OTR-action | PWIL-state | OTR-state | TDGBA |
|---|---|---|---|---|---|
| halfcheetah-medium-v2 | **44.4±0.3** | 43.2±0.2 | 1.1±2.1 | 43.2±0.5 | 42.8±1.4 |
| hopper-medium-v2 | 60.4±5.5 | 74.2±4.8 | 1.4±3.9 | **74.9±1.0** | 58.1±8.2 |
| walker2d-medium-v2 | 78.6±1.3 | 78.7±1.4 | 1.2±2.3 | **79.0±2.2** | 76.5±4.6 |
| halfcheetah-medium-replay-v2 | 42.6±0.9 | 41.8±2.1 | -2.4±1.5 | 41.6±3.3 | **42.7±0.5** |
| hopper-medium-replay-v2 | **94.0±0.6** | 85.4±2.6 | 0.7±1.2 | 84.8±4.4 | 93.3±3.9 |
| walker2d-medium-replay-v2 | 41.9±1.0 | 67.2±1.2 | -0.2±1.2 | 64.5±0.2 | **72.8±3.7** |
| halfcheetah-medium-expert-v2 | 89.5±1.3 | 87.4±1.1 | 0.0±1.5 | 87.6±4.4 | **90.1±1.5** |
| hopper-medium-expert-v2 | 70.9±9.4 | 88.4±3.4 | 2.7±0.8 | 81.1±10.3 | **108.9±2.8** |
| walker2d-medium-expert-v2 | 109.8±0.3 | 109.5±0.7 | 0.2±2.6 | 109.6±0.2 | **109.8±0.4** |
| Average Score | 69.5 | 75.1 | 2.7 | 74.0 | **77.2** |

## 5.5 RESULTS WITH PARAMETER CHANGING

Previous methods typically involve multiple hyperparameters, particularly in reward-based IL methods, where the sensitivity of the offline RL hyperparameters often presents challenges. In contrast, TDGBA requires fewer parameter adjustments and exhibits sufficient robustness to these parameters.

**Scaling coefficient** $\beta$    $\beta$ is the key hyperparameter that determines the alignment measure. We vary $\beta$ across $\{5, 20, 30\}$, and present the final policy performance under varying $\beta$ values in Table 5, which demonstrats sufficient robustness.

**Number of Blocks** $K$    The parameter $K$ determines how many blocks trajectories are divided into after alignment measure calculations. We conducted ablation studies to observe the impact of $K$ on the final policy performance. As shown in Table 6, TDGBA exhibits robustness to variations in $K$.

Table 5: Performances with different $\beta$ values

| Task Name | $\beta = 5$ | $\beta = 20$ | $\beta = 30$ |
|---|---|---|---|
| halfcheetah-medium-replay-v2 | 42.8±1.4 | 42.8±1.1 | 40.5±1.9 |
| hopper-medium-replay-v2 | 93.3±3.9 | 90.4±2.5 | 92.7±2.2 |
| walker2d-medium-replay-v2 | 72.8±3.7 | 70.1±4.5 | 72.6±4.5 |

Table 6: Performances with varying $K$

| Task Name | $K = 10$ | $K = 20$ | $K = 30$ |
|---|---|---|---|
| hopper-medium-replay-v2 | 82.6±3.3 | 93.3±3.9 | 91.2±4.5 |
| antmaze-umaze-v0 | 83.4±4.5 | 90.6±4.2 | 86.8±3.9 |

## 6 DISSUSION

**TDGBA and FTB**    In this section, we provide additional discussion on TDGBA in comparison to FTB(Zhang et al., 2023). To the best of our knowledge, FTB is one of the most state-of-the-art works in the field of offline preference reinforcement learning. Its approach of combining preference learning with trajectory diffusers has demonstrated great potential and has provided significant inspiration for our work. We would like to clarify that although TDGBA adopts methods from FTB for trajectory optimization and policy learning, the core contribution of our work lies in successfully combining the advantages of preference learning with offline imitation learning for the first time. To the best of our knowledge, all previous offline preference learning methods require an additional human preference dataset, whereas our approach can effectively leverage the advantages of preference learning even with only one expert trajectory. Furthermore, it is worth emphasizing that collecting accurate human preference datasets and labeling them, as well as training task-specific preference models, is inherently time-consuming and labor-intensive. TDGBA offers a simpler and more practical advantage in this regard. In the future, we plan to explore combining implicit expert preferences with other preference learning methods and further improving them to continue exploring their value in this field.

**Limitations**    In this section, we highlight some limitations of TDGBA. First, both our approach and most of the baselines are primarily validated on state-based tasks. Applying TDGBA directly to image-based tasks presents challenges, such as high computational costs due to the training of the Trajectory Diffuser and the large dimensionality of images. Additionally, handling multi-modal expert demonstrations may introduce potential complications. To address the above issues, we will explore better alignment measures and investigate more effective preference learning methods to further leverage the advantages of the implicit expert preference concept.

## 7 CONCLUSION

In this paper, we propose Trajectory-level Data Generation with Better Alignment (TDGBA), a novel framework for offline IL without the need for learning behavior weights or reward functions. TDGBA uses the Wasserstein Distance to calculate the alignment measure between unlabeled trajectories and expert demonstrations. Then, TDGBA leverages a small number of expert demonstrations to introduce implicit expert preferences, further guiding the preference-based Trajectory Diffuser to generate high-quality, expert-aligned trajectories. With the generated high-alignment trajectories, we can utilize BC directly to derive the optimal policy. Experimental results on the D4RL benchmarks show that TDGBA significantly outperforms SOTA offline IL methods. Additionally, we demonstrate that incorporating implicit expert preferences effectively combines the advantages of preference learning with scarce expert demonstrations, offering a promising new research direction for the offline imitation learning domain.

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

# A IMPLEMENTATION DETAILS

## A.1 HYPERPARAMETER SETUP

In this section,we detail the hyperparameter setup utilized in TDGBA. We conduct experiments on 9 MuJoCo locomotion "-v2" medium-level dataset and 2 AntMaze "-v0" datasets, yielding a total of 11 tasks. We list the hyperparameter setup for locomotion tasks in Table 7. To acquire expert demonstrations, we use the trajectory with the highest return as expert demonstrations on MuJoCo locomotion tasks and filter the goal-reached trajectory in AntMaze tasks. For all of the baseline offline IL methods, we follow this setting and run them over five different random seeds.We use the normalized score metric recommended in the D4RL paper (Fu et al., 2020), where 0 corresponds to a random policy, and 100 corresponds to an expert policy. Note that we directly use the Wassertein distance as the alignment measure for Antmaze tasks without any scale function.

Table 7: TDGBA hyperparameters for D4RL.

| Hyperparameter | Value |
|---|---|
| **Alignment Measure** | |
| Episode length $T$ | 1000 |
| Cost function | cosine |
| Scale function | $f(d) = \exp\left(5.0 \cdot T \cdot \frac{d}{|A|}\right)$ |
| **Diffusion Model** | |
| Guidance scale $\omega$ | 1.2 |
| Diffusion steps $T$ | 1000 |
| Downsample rate | [1, 2, 4, 8] |
| Hidden dimension | 256 |
| Batch size | 32 |
| Dropout | 0.2 |
| Learning rate | 1e-4 |
| Optimizer | Adam |
| Block number K | 20 |
| Cluster method | KMeans |
| **Policy Learning** | |
| Weight Decay | 2e-4 |

We list the number of unlabeled trajectories $\mathcal{N}$ in all experimental environments and the number of top trajectories $\mathcal{Z}$ chosen for each task below:

Table 8: Number of unlabeled and top trajectories per task

| Task Name | Unlabeled Trajectories $\mathcal{N}$ | Top Trajectories $\mathcal{Z}$ |
|---|---|---|
| halfcheetah-medium-v2 | 1000 | 300 |
| hopper-medium-v2 | 2187 | 300 |
| walker2d-medium-v2 | 1191 | 300 |
| halfcheetah-medium-replay-v2 | 202 | 100 |
| hopper-medium-replay-v2 | 2039 | 300 |
| walker2d-medium-replay-v2 | 1093 | 300 |
| halfcheetah-medium-expert-v2 | 2000 | 300 |
| hopper-medium-expert-v2 | 3214 | 300 |
| walker2d-medium-expert-v2 | 2191 | 300 |
| antmaze-umaze-v0 | 10131 | 50 |
| antmaze-umaze-diverse-v0 | 1035 | 50 |

## A.2 COMPUTATIONAL RESOURCE

For each task, we run our experiments on 5 seeds.We train TDGBA on an RTX 4090, requiring approximately 40 hours, and on an RTX 3090, requiring about 90 hours for one run. The primary time consumption is focused on the training and inference of the diffusion model.

# B OMITTED EXPERIMENTS

## B.1 COMPARISON TO GENERATIVE MODEL IN RL

In this section, we compare our approach with currently popular generative model applications in reinforcement learning (RL). Traditional methods primarily use generative models as planners, predicting sequences of trajectories with high returns and implementing corresponding actions (Chen et al., 2021; Janner et al., 2021; 2022; Ajay et al., 2022). However, these approaches require the ground-truth rewards during training. On the other hand, FTB (Zhang et al., 2023) utilizes an additional human preference dataset to train a task-specific score model, which calculates preferences for unlabeled trajectories. Yet, collecting extensive human preference data is labor-intensive and even challenging. To our knowledge, we are the first to extract optimal strategies using generative models under the conditions of no ground-truth rewards or the dataset with human preference labels. Unlike previous works, we generate high-quality trajectories using only a few expert demonstrations or even a single expert trajectory. We present the performance comparison results with these methods in Table 9, revealing that our approach demonstrates strong competitiveness and highlighting its substantial potential.

Table 9: Comparison to generative model in RL.We report mean $\pm$ standard deviation per task and aggregate performance.

| Task Name | DT | TT | Diffuser | DD | FTB | TDGBA |
|---|---|---|---|---|---|---|
| halfcheetah-medium-v2 | 42.6 | 46.9 | 44.2 | **49.1** | 35.1 | 42.8 $\pm$ 1.4 |
| hopper-medium-v2 | 67.6 | 61.1 | 58.5 | **79.3** | 61.9 | 58.1 $\pm$ 8.2 |
| walker2d-medium-v2 | 74.0 | 79.0 | 79.7 | **82.5** | 79.9 | 76.5 $\pm$ 4.6 |
| halfcheetah-medium-replay-v2 | 36.6 | 41.9 | 42.2 | 39.3 | 39.0 | **42.7 $\pm$ 0.5** |
| hopper-medium-replay-v2 | 82.7 | 91.5 | 96.8 | **100.0** | 90.8 | 93.3 $\pm$ 3.9 |
| walker2d-medium-replay-v2 | 66.6 | 82.6 | 61.2 | 75.0 | **79.9** | 72.8 $\pm$ 3.7 |
| halfcheetah-medium-expert-v2 | 86.8 | **95.0** | 79.3 | 90.6 | 90.6 | 90.1 $\pm$ 1.5 |
| hopper-medium-expert-v2 | 107.6 | 110.7 | 102.6 | **111.8** | 106.9 | 108.9 $\pm$ 2.8 |
| walker2d-medium-expert-v2 | 108.1 | 101.9 | 108.4 | 108.0 | 109.1 | **109.8 $\pm$ 0.4** |
| **Average Score** | 74.7 | 78.9 | 77.3 | 81.8 | 77.0 | 77.2 |

## B.2 COMPARISONS OF GCBC AND BEHAVIOR RETRIEVAL

We add two baselines to compare policy performance with goal-conditioned BC (RvS) Emmons et al. (2021) and Behavior Retrieval Du et al. (2023) on 11 tasks, respectively.

**Experiment Setup:** In order to ensure the fairness of the experiment, we set the recommendation in Emmons et al. (2021) and use "return to go" as the goal condition for Mujoco tasks. For the Antmaze task, we use "target position" as the goal condition. For Behavior Retrieval, we set up our experiments according to the parameters recommended in the paper, where the core parameters filtering threshold $\delta = 0.75$ and $kl\_weight = 0.0001$. We also reproduce the GCBC algorithm according to the parameters recommended in Emmons et al. (2021). We evaluate the performance of the algorithm under 5 random seeds separately, with only one expert trajectory set in each task. The results are shown in Table 10. It can be noticed that TDGBA achieves better performance compared to both methods. The core of the Behavior Retrieval approach is to train a VAE model for calculating the embedding distances of different transitions from the expert transition and picking the closer transition by threshold. In contrast to the previous experimental setup, where there are usually 10

Table 10: Performance comparison with GCBC and Behavior RetrievaL.

| Task Name | GCBC | Behavior Retrieval | TDGBA (ours) |
|---|---|---|---|
| halfcheetah-medium-v2 | 41.3±1.6 | 39.9±0.5 | **42.8 ± 1.4** |
| hopper-medium-v2 | **58.2±3.2** | 55.5±4.3 | 58.1 ± 8.2 |
| walker2d-medium-v2 | 70.9±5.2 | 73.7±1.8 | **76.5 ± 4.6** |
| halfcheetah-medium-replay-v2 | 39.8±0.2 | 16.6±4.2 | **42.7 ± 0.5** |
| hopper-medium-replay-v2 | 76.5±1.7 | 30.3±10.1 | **93.3 ± 3.9** |
| walker2d-medium-replay-v2 | 61.5±1.8 | 20.6±12.5 | **72.8 ± 3.7** |
| halfcheetah-medium-expert-v2 | **92.2±0.8** | 43.2±7.0 | 90.1 ± 1.5 |
| hopper-medium-expert-v2 | 100.0±1.1 | 72.5±1.8 | **108.9 ± 2.8** |
| walker2d-medium-expert-v2 | 105.4±0.6 | 105.2±0.3 | **109.8 ± 0.4** |
| antmaze-umaze-v0 | 70.1±2.6 | 55.6±9.9 | **90.6 ± 4.2** |
| antmaze-umaze-diverse-v0 | 62.7±8.4 | 44.9±10.3 | **72.3 ± 6.5** |

expert trajectories and nearly 50% of expert trajectories in the unlabeled dataset. However, in our experimental setup, only one expert trajectory is used, and datasets like "medium" and "medium-replay" types contain less expert data and numerous inferior trajectories. This makes it difficult to learn an accurate VAE model, which may lead to the failure to accurately identify high-quality transitions. On the contrary, TDGBA can filter out high-quality trajectories and further optimize them even when there is only one expert trajectory, demonstrating the strong potential of our method.

### B.3 COMPARISONS OF REWARD LEARNING AND ALIGNMENT MEASURE

In this section, we present the density differences between the rewards learned by reward learning algorithms and the ground-truth rewards across multiple tasks, as illustrated in Figure 5. Additionally, for these tasks, we further demonstrate the correlation between alignment measures and ground-truth returns in Figure 6.

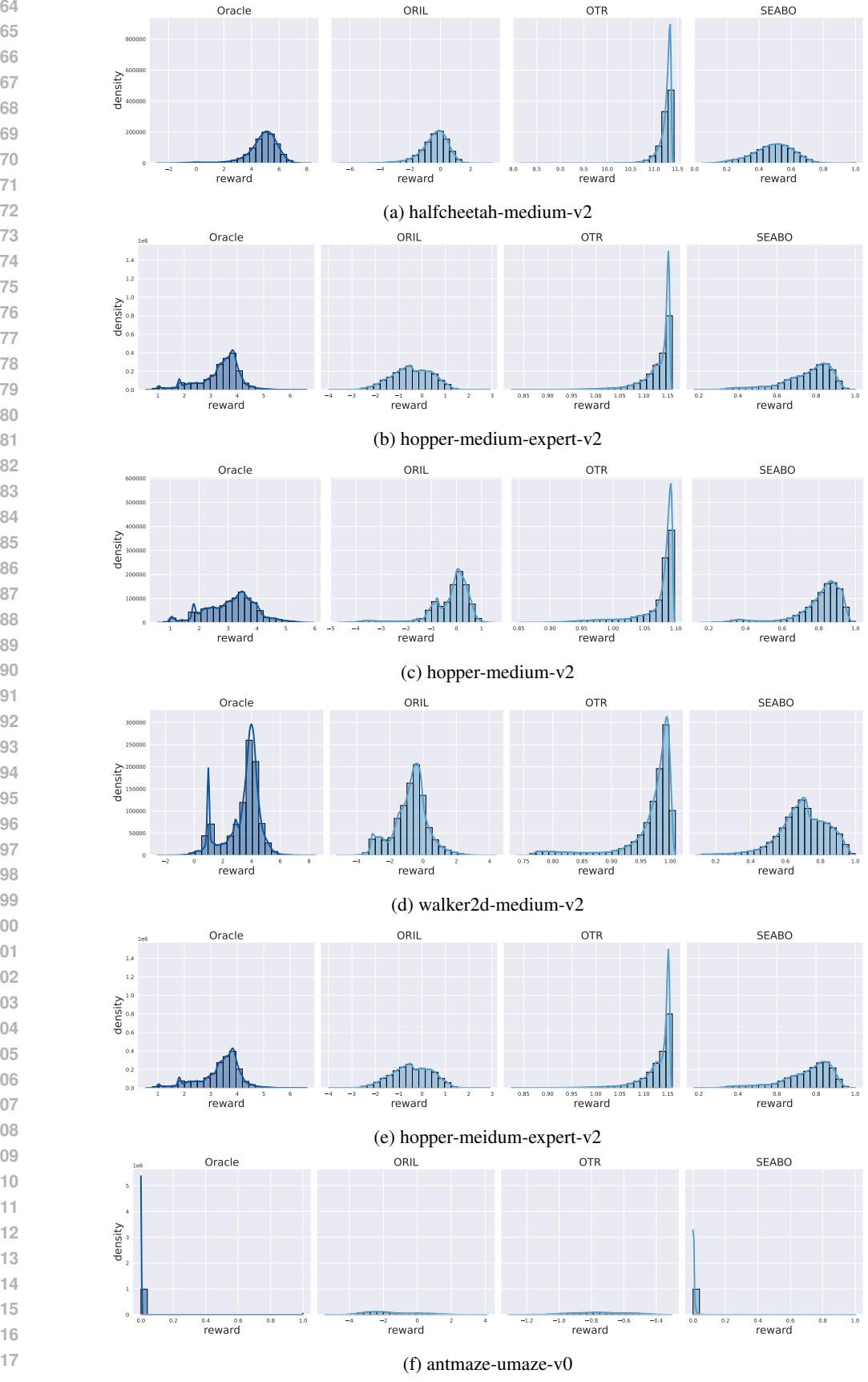

Figure 5: Density plots of ground-truth rewards and learned rewards.

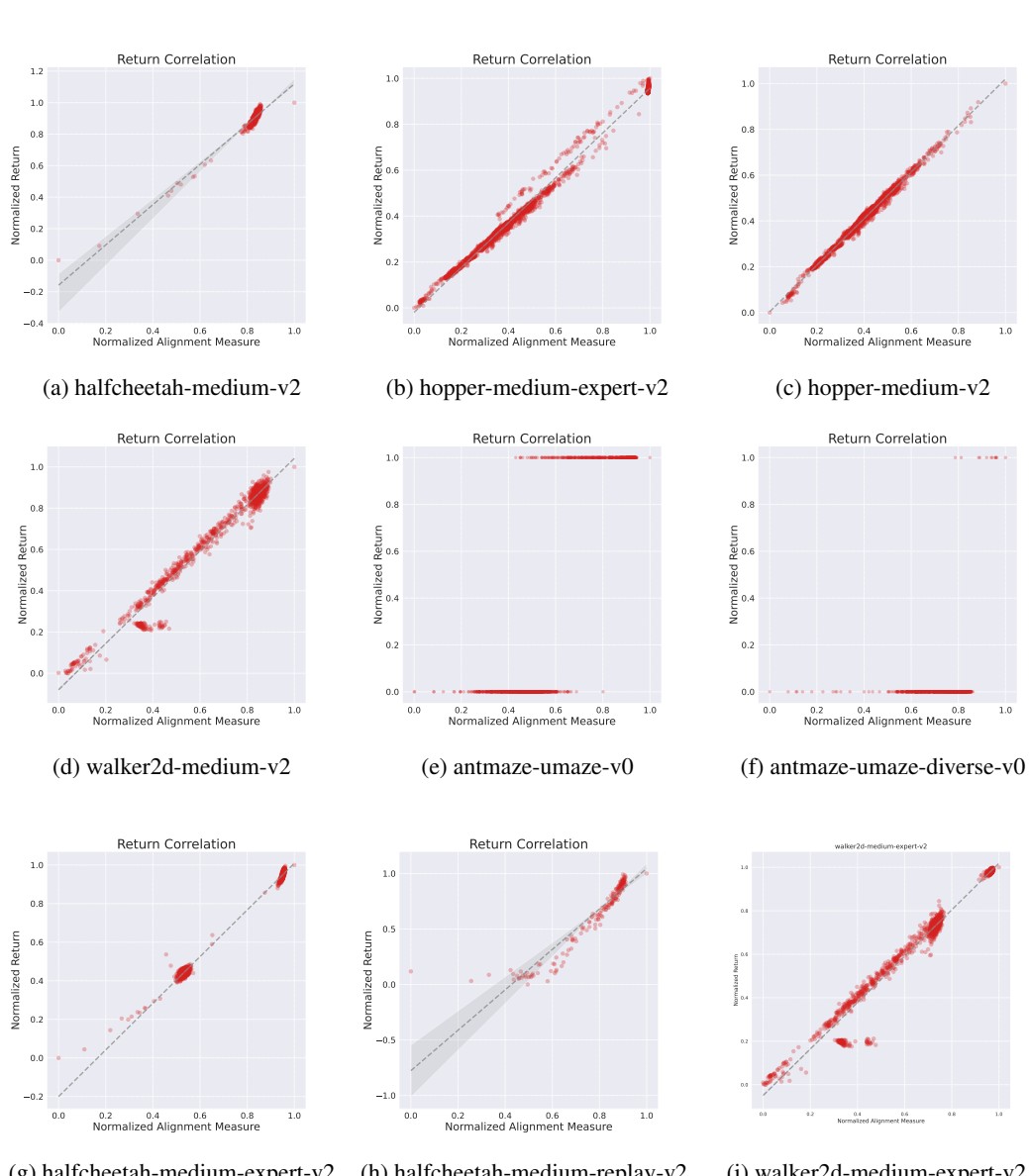

Figure 6: Correlation between alignment measures and ground-truth returns.

