# OpenReview forum: "Trajectory-level Data Generation with Better Alignment for Offline Imitation Learning"
_ICLR.cc/2025/Conference — Submitted to ICLR 2025_

### Official Review · Reviewer_ht2y · 2024-10-22

**Soundness:** 2
**Presentation:** 3
**Contribution:** 2
**Rating:** 5
**Confidence:** 4

**Summary:**

It introduces Trajectory-level Data Generation with Better Alignment (TDGBA), which addresses the challenges of scarce expert data and suboptimal trajectories by using a diffusion model guided by alignment measures between unlabeled trajectories and expert demonstrations. The method employs an alignment metric based on optimal transport theory and Wasserstein distance, which is used as an implicit expert preference to train the diffusion model. This approach generates high-quality, diverse trajectories that enable the direct application of Behavior Cloning methods to derive optimal policies, outperforming state-of-the-art offline IL methods on D4RL benchmarks⁠.

**Strengths:**

This paper introduces a metric using Wasserstein distance to measure the alignment between unlabeled trajectories and expert demonstrations.

**Weaknesses:**

The conditional diffusion model conditioned on inferior data to generate better data is derived from previous work [1]. Therefore, it is essentially an adaptation to the offline imitation setting.

The defined metric has some practical limitations. Perfect demonstrations may be scarce, and the policy might be multi-modal, making it challenging to formally define the distance. However, practical methods to measure differences, such as training a discriminator, have been proposed in previous work [2].
Thus, the contribution of this work appears limited.

[1] Zhang Z, Sun Y, Ye J, et al. Flow to better: Offline preference-based reinforcement learning via preferred trajectory generation.

[2] Wang, Yunke, Chang Xu, and Bo Du. Robust Adversarial Imitation Learning via Adaptively-Selected Demonstrations.

**Questions:**

Can you visualize the improved generated data? Given that the original conditional diffusion might rely on a large number of human labels, does the expert-demonstration-shaped preference satisfy the data requirements? Ablation studies on the number of preference labels and trajectories are also needed.

---

> ### Author Response · Authors · 2024-11-25
> **Official Comment by Authors**
>
> Dear Reviewer,
>
> We are very grateful for the time you spent reviewing our work and for the valuable suggestions you provided. We hope the following content will provide a better understanding of our work.
>
> **[Weakness 1]**
>
> We thank the reviewer for pointing out this weakness. We would like to clarify that the primary goal of our work is to introduce the concept of implicit expert preferences and to attempt to effectively transfer the advantages of preference learning to offline imitation learning. Notably, in the field of offline imitation learning, we only have a limited number of expert demonstrations and a large number of unlabeled trajectories. Due to the lack of prior human preference datasets, previous works have been unable to fully leverage the benefits of preference learning. Considering that the core of our work is to explore a new and promising research direction, we have thoroughly investigated the current methods in the field of offline preference reinforcement learning. We found that [1] achieves state-of-the-art performance in multiple tasks, and we have adopted their approach to generating data using a conditional diffusion model in our work. In the future, we will optimize the generative model or explore other methods to further enhance the advantages of implicit expert preferences.
>
> **[Weakness 2]**
>
> We greatly appreciate the reviewer's suggestions and recommended works. As mentioned earlier, our work is a new research attempt to integrate the advantages of preference learning into offline imitation learning, with a focus on effectively transferring preference learning to the offline imitation learning setting. Currently, we use only one expert trajectory in our experiments to maintain the scarcity of expert demonstrations. Regarding issues such as inaccurate distances due to the multimodality of policies, we will conduct further research in the future. We hope to further explain the contributions of our method here. Previous preference learning works were limited by the need for additional human preference datasets, which require significant guarantees in constructing these datasets. For instance, the differences between pairs of preference trajectories should not be too large; different humans may have varying degrees of preference for trajectories, potentially leading to multimodality or contamination in the dataset. In contrast, obtaining one or a few expert trajectories is more feasible, allowing us to effectively leverage the advantages of preference learning while avoiding the time-consuming and labor-intensive process of constructing preference datasets.
>
> **[Question 1]**
>
> Currently, we have not rendered the generated data in our work. We will supplement the visualization results of this content in the revised version.
>
> **[Question 2]**
>
> In Figures 3 and 6 of our work, we have visualized the correlation between implicit expert preferences and ground-truth returns. It can be observed that even with only one expert trajectory, this metric can accurately capture the differences between different trajectories and the expert trajectory, showing a strong correlation. This allows us to distinguish between good and bad trajectories and construct accurate preference trajectory pairs.
>
> **[Question 3]**
>
> We are slightly unclear about this question. We believe you are asking us to perform ablation studies on the trajectories in the dataset under different quantities and qualities. First, we would like to clarify that for the same task, such as Hopper, the D4RL[2] dataset typically provides three types of datasets: 'medium-play', 'medium-expert', and 'medium'. The trajectories in these datasets are collected by policies with different performance levels, inherently differing in both quantity and quality. For example, 'medium-expert' usually contains more high-return trajectories, and policies learned from this dataset typically achieve higher performance compared to the 'medium' setting. The specific number of trajectories under different settings can be found in Table 7.
>
> Thank you.
>
> [Reference]
>
> [1] Flow to better: Offline preference-based reinforcement learning via preferred trajectory generation. ICLR, 2024.
>
> [2] D4RL: Datasets for Deep Data-Driven Reinforcement Learning. 2021.

---

### Official Review · Reviewer_yrmB · 2024-10-23

**Soundness:** 3
**Presentation:** 3
**Contribution:** 2
**Rating:** 3
**Confidence:** 1

**Summary:**

This paper proposes a simple yet effective trajectory alignment approach that combines a W1-distance-based trajectory ranking method with the use of a diffusion policy. This approach demonstrates improved performance in the imitation learning (IL) domain, specifically within Gym-Mujoco.

Additionally, the authors provide extensive analytical experiments and analysis, which illustrate the limitations of previous methods' reward shaping. These experimental results are crucial for understanding the performance of current reward-shaping algorithms.

However, the extent to which improvements result from algorithm innovation rather than model architecture still needs further verification.

**Strengths:**

This paper proposes a simple yet effective trajectory alignment approach that combines a W1-distance-based trajectory ranking method with a diffusion policy. It demonstrates improved performance in imitation learning (IL) tasks sourced from the Gym-Mujoco domain.

Additionally, the authors conducted extensive analytical experiments and analyses, which highlight the limitations of previous reward shaping methods. These experimental results are essential in helping us understand the constraints of current reward shaping-based IL algorithms.

**Weaknesses:**

See Question Section

**Questions:**

**Q.1** How much improvement is brought by your trajectories ranking and sorted out? since the performance of Diffusion policy is considerate enoughy, especially in the long horizonal setting. Meanwhile, most of your baselines chosen are based on MLP policies.

**Q.2**  From your experimental results, I observe that most of your setups focus on continuous control tasks, while there is a lack of long-horizon decision-making tasks. Could you please provide more comparisons across these related domains (such as kitchen, android, etc.)? Thank you.

**Q.3** Could you clarify the advantages of your paper over other alignment and sequential modeling approaches? For instance, the paper [1] utilizes contextualized information to align demonstrations, showcasing improved performance on IL tasks. I also have a question about your **diffusion policy's optimizing objective**.

Specifically, we know diffusion policy can be optimized via ODE or SDE based method. ODE objective includes consistency model e.g.  and SDE methods encompass numerous related researches. Most of those methods optimize models via adding noising to the initial feature till uniform distribution followed by recovering the feature or predicting the added noise. Therefore, I am very confused about your diffusion objective. Why its a maximizing likelihood objective?

If you ignore your objective, I think it's not diffusion policy, I prefer naming it U-NET? Meanwhile, I can provide some method for you to check whether its diffusion policy:

- Render and observe, whether your policy can learn multiple modes. For example, given a state, there are several clusters.
- check your codebase, and observe, whether you just don't correctly write the objective.

I won't directly reject this paper, since your contribution is independent with diffusion policy. I am looking forward to any improvements or corrections.

Reference
[1] Z. Zhang, J. Xu, J. Liu, Z. Zhuang, D. Wang, M. Liu, S. Zhang, Context-Former: Stitching via Latent Conditioned Sequence Modeling

---

> ### Author Response · Authors · 2024-11-25
> **Official Comment by Authors (1)**
>
> Dear Reviewer,
>
> Thank you for taking the time to review our manuscript and for raising many valuable questions. We will address your concerns in detail below.
>
> **[Question 1]**
>
> In response to the reviewer's query regarding the Diffusion Policy method, we have reviewed the corresponding literature [1] and identified some inconsistencies in the background settings. Firstly, Diffusion Policy is a powerful paradigm for imitation learning that has emerged in recent years. Its capability to model multimodal action sequence distributions can effectively mitigate the compounding error problem inherent in traditional MLP policy methods, achieving excellent performance in robotic manipulation tasks. However, a significant difference lies in the requirement for Diffusion Policy to learn from a sufficiently clean dataset with a considerable amount of expert trajectories. When the dataset includes suboptimal or even failed trajectories, it may pose issues for the final imitated policy. In our setting for offline imitation learning, we only have a small number, or even a single expert trajectory, along with a large amount of unlabeled suboptimal trajectories. Under such conditions, directly learning a diffusion policy would struggle to cover the entire state space, which is why we did not conduct a direct performance comparison with Diffusion Policy. Thus, one of the key challenges our work addresses is how to find an effective metric to distinguish high-quality trajectories within an unlabeled dataset. Moreover, we aim to make full use of suboptimal and failed trajectories by transferring the advantages of preference learning into the domain of offline imitation learning. Specifically, we opted for the state-of-the-art conditional diffusion model-based trajectory optimization approach in the offline preference RL field. It is worth noting that the core of our work is the concept of discovering implicit expert preferences, and how to better utilize diffusion models or other policy models will be explored in future research.
>
> **[Question 2]**
>
> We appreciate the insightful comments provided by the reviewer. As you pointed out, our current method focuses on continuous control tasks. During experimentation with other environments (e.g., Adroit and Kitchen), we found that our method offers limited performance improvements, and there are even greater difficulties in training the conditional diffusion model to effectively augment trajectories. The underlying reasons, as we analyze, might be due to potential issues in calculating implicit expert preferences for long-horizon decision-making tasks, as well as the increased state dimensions and trajectory lengths making the current model less suitable. Future work will involve further exploration in these areas.
>
> **[Question 3]**
>
> We are deeply grateful for the reviewer's valuable suggestions. We carefully read the paper on Context-Former, which we find to be an intriguing contribution that has greatly inspired us, despite the authors not having released the related code at present. This work presents a distinct solution compared to our approach, it primarily addresses concatenation and OOD (Out-of-Distribution) problems inherent in Decision Transformer-like methods. By employing the key idea of Hindsight Information Matching, the authors use the optimal encoding representation $ z^*$ of expert trajectories as a condition, replacing the return-to-go used in Decision Transformers, thereby aligning the generated trajectory sequences with the optimal trajectory representations. These segments can be sampled from expert-like portions of suboptimal trajectories, partly resolving the concatenation issue. During training, an additional BERT-style HIM extractor $I_{\phi}$ is introduced to obtain the expert representation. Based on the above description, our method initially follows the trajectory generation approach in FTB [2] and employs a simple BC policy in the policy network, which results in faster inference speeds compared to DT-style methods but may also lead to more compounding errors during inference. Additionally, we observe that the work under discussion uses between $5$ to $20$ expert trajectories in experiments, and it remains unclear whether their method can handle the extreme case of having only a single expert trajectory. To summarize, our method integrates the benefits of preference learning into the offline imitation learning domain, while Context-Former improves policy performance from the perspective of expert matching. Both approaches offer excellent solutions to the offline IL domain, and we plan to attempt combining the strengths of both methods and perform more detailed comparisons in future work.

---

> ### Author Response · Authors · 2024-11-25
> **Official Comment by Authors (2)**
>
> Regarding the reviewer's question about the 'maximizing likelihood objective' , whether using diffusion models, VAEs, or GANs, our goal is to model the conditional probability distribution $p(x|y)$, where low-preference trajectories serve as the condition and high-preference trajectories as the target distribution. The choice of diffusion models is mainly attributed to their stability and strong ability to fit complex distributions. The processes involving ODE, SDE, and noise procedures are part of the implementation of diffusion models, including the noise prediction model, etc.
>
> We hope these clarifications are helpful, and we look forward to any further feedback you may have.
>
> Thank you.
>
> [Reference]
>
> [1] Diffusion policy: Visuomotor policy learning via action diffusion. RSS, 2023.
>
> [2] Flow to better: Offline preference-based reinforcement learning via preferred trajectory generation. ICLR, 2024.

---

> ### Comment · Reviewer_yrmB · 2024-11-25
> **Further concern**
>
> As you said, diffusion can fit complex distributions. Is this due to the diffusion objective function or the model? If your policy doesn't have the characteristics of a diffusion policy, is it still a diffusion policy? Of course, I should mention that this has nothing to do with your contribution.
>
> Please review the papers related to diffusion policy. These papers, through the rendered policy-generated distributions, can intuitively make readers aware that trained models can fit nonlinear distributions. Even the simplest paper target the inverse process of adding noise as the optimization goal, rather than simply maximizing likelihood.

---

> > ### Comment · Reviewer_yhxs · 2024-11-25
> >
> > Dear Reviewer yrmB,
> >
> > I believe their diffusion model is trained using Equation 5, which represents a standard denoising score-matching objective. I assume the authors introduced Equation 4 first because the diffusion training objective is actually a lower bound on the logarithm of the likelihood, $\log p(x)$ (refer to Theorem 2.6 and 2.7 in [1]). However, I agree that their explanation for transitioning from Equation 4 to Equation 5 seems unclear or insufficiently detailed.
> >
> > [1] Chan, S. H. (2024). Tutorial on Diffusion Models for Imaging and Vision. arXiv preprint arXiv:2403.18103.

---

> > > ### Comment · Reviewer_yrmB · 2024-11-25
> > > **Thanks for Reviewer yhxs' Comment**
> > >
> > > Thank you for helping the author clarify this point. However, the author uses the maximum likelihood objective, which is Equation (4), in the tenth line of the pseudocode. As you mentioned, Equation (5) is the standard optimization objective for diffusion models, but the connection between Equation (4) and Equation (5) has not been clarified.

---

> > > > ### Comment · Reviewer_yhxs · 2024-11-25
> > > >
> > > > I agree. Line 10 in the pseudocode is quite weird.

---

> ### Author Response · Authors · 2024-12-03
>
> Dear Reviewer,
>
> We would like to express our gratitude for your careful review of our work.
>
> In the updated version of the manuscript, we have provided a more detailed expansion of the "Flow-to-Better with Trajectory Diffuser" in the "Preliminaries" section. Additionally, we have removed the pseudocode and clarified the previously ambiguous areas in the methodology section.
>
> Once again, we sincerely appreciate the time and effort you have dedicated to reviewing our work.
>
> Best regards,
>
> All authors

---

> ### Comment · Reviewer_yrmB · 2024-12-03
> **Official Comment by Reviewer yrmB**
>
> Thank you for confirming that it is a diffusion policy. Furthermore, your paper still lacks a **very important** ablation experiment that **maintains all models as diffusion-based and compares the performance of the TDGBA and diffusion combined with DICE, IL methods**.  However, I believe this is a relatively challenging problem in engineering. Therefore, you can compare different diffusion policies on the same dataset to determine whether it is your algorithm that brings improvements to the diffusion process.
>
> Meanwhile, the issue discussed by you and the reviewer oZ9p is, in my opinion, one that has the greatest impact on this paper.
>
> Therefore, I have decided to lower the score:
>
> - Reason 1: I have already mentioned this issue in the first round of review comments, but I haven't seen any action taken in the past two weeks.
>
> - Reason 2: I believe your comments with Reviewer oZ9p  will be a significant factor influencing this paper, as your current writing is already relatively improved, and this can be continuously revised, and with the only shortcoming being insufficient experiments. However, Reviewer oZ9p's concerns have a relatively large impact.
>
> While, I lower down the confidence to 1, leaving it to the AC to judge whether to accept the paper based on the opinions of other reviewers while certainly take my mentioned ablation experiment into consideration.

---

> > ### Author Response · Authors · 2024-12-04
> >
> > Dear Reviewer,
> >
> > We sincerely appreciate your constructive feedback and the time you have invested in reviewing our work. We would like to provide further clarification regarding the experimental concerns related to the diffusion policy that you raised. It appears that there was some misunderstanding between us in this regard during our previous exchanges.
> >
> > Firstly, diffusion policy itself is a decision-making model designed to model the multimodal distribution of actions or action sequences, $p(a|s)$ , with the model outputting actions that the agent takes in interaction with the environment. In contrast, the Trajectory Diffuser, which we and FTB utilize, models the distribution between preferred trajectories, $p(\tau^h|\tau^l)$ , and can be regarded as a data augmentation model aimed at generating high-quality data for training decision-making models.
> >
> > In terms of the decision-making model, we continue to employ a simple behavior cloning method, using an MLP network architecture, to learn the mapping from states to actions. Thus, in terms of decision-making, we align with approaches such as DICE. The key difference is that, within the offline training dataset, we have employed the diffusion model for trajectory optimization and data augmentation.
> >
> > Regardless of your final evaluation, we once again extend our sincere thanks for the time and effort you have dedicated to reviewing our work.  We wish you all the best in your future research and professional endeavors.
> >
> > Best regards,
> >
> > All authors

---

### Official Review · Reviewer_D3TZ · 2024-10-29

**Soundness:** 2
**Presentation:** 3
**Contribution:** 3
**Rating:** 6
**Confidence:** 2

**Summary:**

Offline reinforcement learning depends on precise reward signals, which are difficult to obtain. Offline imitation learning (IL) seeks to develop policies from expert demonstrations without rewards, but is limited by scarce expert data and numerous suboptimal trajectories, affecting methods like behavior cloning (BC). Traditional approaches using importance weights or reward functions for BC encounter instability and accuracy issues. To address this, we introduce Trajectory-level Data Generation with Better Alignment (TDGBA). This method aligns unlabeled trajectories with expert demonstrations to guide a diffusion model in generating well-aligned trajectories, enabling BC to extract optimal policies directly. It also uses implicit expert preferences to improve stability, fidelity, and diversity. Experiments on D4RL benchmarks demonstrate TDGBA's superior performance over other offline IL methods, confirming the effectiveness of diffusion models and expert preferences in trajectory data generation.

**Strengths:**

well-written and clear

well-motivated

extensive comparisons and evaluations in experiments

**Weaknesses:**

The grammar and presentation of some paragraphs need to be improved.

The visualization of the experimental results needs to be improved slightly.

**Questions:**

line 212, why using lower-alignment trajectories as the condition？

In Section 4.3, the visualization results were only presented on the hopper-medium-replay-v2 task, which left the reader somewhat confused and curious. How about the results on other tasks?

---

> ### Author Response · Authors · 2024-11-24
> **Official Comment by Authors**
>
> Dear Reviewer,
>
> We are very grateful for the time you spent reviewing our work, and we hope the following responses will provide a better understanding of our research.
>
> **[Weakness]**
>
> We appreciate your suggestions regarding grammatical expression and result visualization. We will re-examine our work in these areas and make appropriate revisions.
>
> **[Question 1]**
>
> Thank you for raising this question. The primary purpose of using low-alignment trajectories as conditions is to model the conditional distribution $ p(\tau^h|\tau^l) $, where $ \tau^l $ represents low-alignment trajectories and $ \tau^h $ represents high-alignment trajectories. The diffusion model trained based on this conditional distribution incorporates the idea of preference learning, which can gradually optimize low-preference trajectories into high-preference ones. This modeling approach has been proposed and validated in previous work [1].
>
> **[Question 2]**
>
> Thank you for your suggestion. We will supplement the visualization results for other tasks in the revised version to avoid any potential confusion.
>
> Thank you.
>
> [Reference]
>
> [1] Flow to better: Offline preference-based reinforcement learning via preferred trajectory generation. ICLR, 2024.

---

> > ### Comment · Reviewer_D3TZ · 2024-11-25
> >
> > Thanks for your early reply. You have addressed all my concerns, and I have increased the score from 5 to 6. Regardless of what other reviewers commented, I think your work has done a good job. No matter this paper is rejected or not finally, please keep going on in the future.

---

> > > ### Author Response · Authors · 2024-11-29
> > >
> > > Thank you very much for your response and support for our work. It has deeply encouraged us to keep striving forward. Wishing you all the best in your future endeavors!

---

### Official Review · Reviewer_yhxs · 2024-10-31

**Soundness:** 2
**Presentation:** 3
**Contribution:** 2
**Rating:** 3
**Confidence:** 3

**Summary:**

The paper presents a novel offline imitation learning method, Trajectory-level Data Generation with Better Alignment (TDGBA), designed to optimize policy learning when both expert and unlabeled datasets are available. TDGBA leverages the Wasserstein Distance to measure alignment between unlabeled and expert trajectories, constructing preference pairs from the initially unlabeled dataset. A trajectory diffusion model is then trained to generate higher-aligned trajectories conditioned on lower-aligned ones, progressively enhancing the top-aligned trajectories. The behavior cloning (BC) policy trained on this improved dataset demonstrates significant improvement over baseline algorithms, particularly in mixed-quality datasets such as medium-expert and medium-replay.

**Strengths:**

1. TDGBA uses implicit preference labeling to guide trajectory generation, enhancing the quality of generated trajectories without requiring additional human labels or preference datasets.

2. TDGBA avoids the complexities and computational costs associated with reward learning in offline settings, simplifying the imitation learning process.

3. Experimental results show that TDGBA outperforms state-of-the-art offline IL methods, especially in mixed-quality datasets like medium-expert and medium-replay.

**Weaknesses:**

1. Some critical algorithm designs lack sufficient and reasonable explanations. 1) Why are low-preference trajectories used as conditions of the classifier-free guidance, and how about directly using labeled preference values as generation conditions? 2) According to Algorithm 1, the expert demonstrations were not used as training data for BC policy. What is the reason for this choice?

2. Some important technical details are missing on the paper. As the authors did not provide their code implementation, reproducing the results could be challenging. Specifically, what is the length of the trajectories generated by the diffusion model? If it corresponds to the length of an episode, e.g., 1000 steps in MuJoCo, would this make the model’s inference cost very high? Also, the authors did not mention the model architecture of the diffusion model. Even though the downsample rate in the parameter table suggests it might be a U-Net, the authors did not explicitly clarify this in the paper.

3. (minor issues) In Table 6, the "Top trajectories number" is reported as 300, but in Table 7, it varies by environment. Line 47 is missing a space before the left parenthesis, line 197 is missing a closing parenthesis, “confirme” on line 268 should be corrected to "confirm", and a space is missing after the closing parenthesis on line 318.

**Questions:**

1. In line 3 of Algorithm 1, how did the authors select the expert demonstration trajectory to pair with each unlabeled trajectory? If the selection was random, could this pairing approach create issues in environments where the initial state varies significantly, as the optimal trajectory distribution may differ widely depending on the initial state?

2. Since the diffusion model is applied solely to improve the top trajectories, is it necessary to train on all neighbor blocks? For instance, training the model to generate $B_2$ from $B_1$ might be unnecessary.

---

> ### Author Response · Authors · 2024-11-24
> **Official Comment by Authors**
>
> Dear Reviewer,
>
> We sincerely thank the reviewer for the constructive feedback and comments. Such a thoughtful review can significantly enhance the quality of our work. We will address each of your concerns and questions in detail:
>
> **[Weakness 1]**
>
> W1.1: In our current work, we utilize low-preference trajectories as guiding conditions because we have found that this approach performs well in related offline preference reinforcement learning tasks [1]. We believe that using labeled preference values is similar to the generation methods used in Decision Diffuser [2] and Decision Transformer [3]. Since our focus in this work is on how to construct implicit expert preferences using a small number of expert demonstrations, thereby leveraging the advantages of preference learning in offline imitation learning, we will conduct more experimental analyses and optimizations on the design of conditional diffusion models in future work.
>
> W1.2: During our experiments, we currently use only one expert trajectory as a demonstration. Although it is not directly included in the training data, the expert demonstration in our experiments originates from the offline dataset and is typically selected based on the actual trajectory rewards of the environment. According to our alignment metric, the corresponding implicit expert preference is the highest, so it is passively included in the final BC policy dataset.
>
> **[Weakness 2]**
>
> W2.1: As you rightly pointed out, we have chosen a trajectory length of $ 1000 $ in the MuJoCo environments. The current diffusion model requires denoising the entire trajectory during training, which indeed incurs a significant computational cost. We will attempt to optimize the design of the diffusion model in subsequent work to reduce computational overhead. Furthermore, we are providing an anonymous link to the code: https://anonymous.4open.science/r/TDGBA-F256/README.md. We plan to release the full code once this work is accepted.
>
> W2.2: We appreciate your valuable suggestions, which can make our work more comprehensive. Given that our focus is on proposing the idea of implicit expert preferences, we chose the powerful Trajectory Diffuser[1] currently used in offline preference reinforcement learning. We will supplement the detailed model structure description in the revised version of our work.
>
> **[Weakness 3]**
>
> Thank you for pointing out the discrepancies in the 'top trajectories number' values presented in Tables 6 and 7. We will remove this item from Table 6 in the revised version to avoid any potential ambiguity and emphasize the differences in the number of trajectories selected across different environments in Table 7. Regarding the errors in symbols and words you mentioned, we will make targeted corrections.
>
> **[Question 1]**
>
> Thank you for your meticulous review and insightful questions. Firstly, in our current experiments, we have chosen only one expert trajectory as the expert demonstration trajectory. This aligns with common practices in the field of offline imitation learning, where the D4RL [4] offline datasets provided include ground-truth environmental rewards. We calculate the cumulative reward for each trajectory based on these rewards and select the trajectory with the highest return as the expert demonstration.
>
> **[Question 2]**
>
> We appreciate your valuable suggestions once again. As previously mentioned, the current focus of our work is not on the design and training of diffusion models but rather on proposing the concept of implicit expert preferences to effectively transfer offline preference learning methods to offline imitation learning. We hope this work can inspire more researchers to explore this area. In the future, we will attempt to design and optimize various diffusion model schemes.
>
> Thank you.
>
> [Reference]
>
> [1] Flow to better: Offline preference-based reinforcement learning via preferred trajectory generation. ICLR, 2024.
>
> [2] Decision transformer: Reinforcement learning via sequence modeling. NIPS, 2021.
>
> [3] Diffusion Policies as an Expressive Policy Class for Offline Reinforcement Learning. ICLR, 2023.
>
> [4] D4RL: Datasets for Deep Data-Driven Reinforcement Learning. 2021.

---

> > ### Comment · Reviewer_yhxs · 2024-11-27
> >
> > I appreciate the authors for their responses, which addressed several of my initial concerns. However, this submission still has notable weaknesses, including limited novelty (as highlighted by Reviewer oZ9p), poor scalability and high computational cost (W2.1), and the reliance on a single expert trajectory. The latter may rest on strong assumptions regarding the initial state distribution and minimal stochasticity in the environment dynamics. Therefore, I stand by my initial rating.

---

> > > ### Author Response · Authors · 2024-12-03
> > >
> > > Dear Reviewer,
> > >
> > > We sincerely appreciate the time and effort you have dedicated to providing feedback.
> > >
> > > Regarding your concerns about scalability, we plan to explore combining implicit expert preferences with other preference learning methods in future work, to further demonstrate the effectiveness and scalability of such preferences. Additionally, the choice of using a single expert trajectory in this study was twofold: on one hand, to emphasize the advantages of the method in scenarios with extremely scarce expert trajectories; on the other hand, for multiple expert trajectories, we can employ clustering and minimum distance techniques to construct preference datasets for different trajectories within different clusters, thus mitigating potential challenges arising from the multimodal nature of expert trajectories.
> > >
> > > Once again, we thank you for your valuable suggestions.
> > >
> > > Best regards,
> > >
> > > All authors

---

### Official Review · Reviewer_oZ9p · 2024-11-03

**Soundness:** 1
**Presentation:** 2
**Contribution:** 1
**Rating:** 1
**Confidence:** 5

**Summary:**

This paper proposes a method called Trajectory-level Data Generation with Better Alignment (TDGBA)  for offline imitation learning (IL). TDGBA leverages alignment measures between unlabeled trajectories and expert demonstrations to guide a diffusion model in generating highly aligned trajectories, which are then used for better behavior cloning. Experimental results on the D4RL demonstrate that TDGBA outperforms SOTA IL methods.

**Strengths:**

N/A

**Weaknesses:**

I have sufficient reason to believe that this paper has plagiarized another paper [1] (called flow-to-better (FTB)) in both its method and writing. I will explain this in detail:

**Method:**

The method proposed in this paper, TDGBA, first uses an alignment measurement method to score the trajectories in the dataset, then applies a clustering method to divide all trajectories into several blocks. Next, two trajectories from two neighboring blocks are sampled, respectively labeled as high-alignment and low-alignment trajectories, and these are provided to a diffusion model for learning. Finally, the learned diffusion model is used to generate several high-alignment trajectories as augmented data for imitation learning. This entire process is identical to the FTB method, except that in FTB, human preferences are used as the measurement in the first step, whereas TDGBA proposes its own alignment measurement. However, the paper completely avoids discussing these similarities, only mentioning “Drawing inspiration from previous works...” in Section 3.2.

**Writing:**

The overall structure of the paper is also highly similar to the FTB paper. For example, Section 3.2 is very similar to Section 3.1 in FTB. The experimental sections 4.2 and 4.3 are also similar to sections 4.2 and 4.3 in FTB, and the discussion of generative models in the related work section also closely resembles that of FTB.

In summary, while TDGBA and FTB are methods proposed in different fields—one in offline imitation learning and the other in offline preference-based RL—their method structure, details, and even parts of the writing are very similar. Therefore, I have reason to suspect that this paper may have plagiarized from FTB.

[1] Zhang et al. "Flow to Better: Offline Preference-based Reinforcement Learning via Preferred Trajectory Generation" (ICLR'24).

**Questions:**

I have no further questions because this paper is suspected of plagiarism.

---

> ### Author Response · Authors · 2024-11-15
> **Official Comment by Authors (1)**
>
> Dear reviewer:
>
> We sincerely thank the reviewer for the time and effort in reviewing our work. We fully understand and respect your role as a reviewer in ensuring the quality of the work, but **we must clarify that we totally disagree with the conclusion regarding plagiarism.** **We assure that we have never had any intention of plagiarizing other works. We will make every effort to provide a detailed explanation to address your concerns, as this is crucial for both our current work and future endeavors.**
>
> **[Method]**
>
> First, we would like to provide a clearer explanation of the original motivation behind the proposed TDGBA method. As you noted, TDGBA is applied to the field of offline imitation learning, where **a key challenge is the lack of labeled data to distinguish good from bad trajectories, with only a small number of expert demonstrations, or even a single expert trajectory, being provided.** Through our extensive review of the literature, we found that previous works have mainly focused on inverse reinforcement learning or methods from the DICE family. However, these methods are often limited by the finite state space in the training set or inaccurate reward estimates. As a result, when encountering unseen states during inference, they tend to suffer from compound errors in the policy. **In response to this issue, we aim to address the problem from the perspective of data optimization and augmentation.**
>
> **To the best of our knowledge, there has been no prior work in offline imitation learning that directly tackles this challenge using data generation.** The primary reason for this gap is the scarcity of expert demonstrations and the difficulty in utilizing large volumes of unlabeled data. **We are grateful for the inspiration provided by the FTB method from offline preference-based RL,  particularly its use of human-preference learning in diffusion model to optimize trajectories.** However, the FTB approach requires an additional human preference dataset to train a task-specific score model, which is used to label each trajectory. We believe that collecting such human preference datasets could present several challenges: 1) the differences between preferences should not be too large; 2) individuals may have varying preferences for the same trajectories; 3) as task complexity increases or the collection of human preferences becomes more difficult, the accuracy of the score model may decrease, potentially impacting the quality of subsequent labeling.
>
> **Therefore, under the setting of offline imitation learning where only a limited number of expert trajectories are available, a key aspect of our method is to find an effective metric to label different trajectories.** This allows us to construct a form of preference that does not rely on human input, which we refer to as "implict expert preference" in our work. We believe that by applying the concept of preference learning in this context—without relying on human preference datasets—we can effectively tackle several challenges inherent in offline imitation learning. **This is a central focus of our work and represents a promising direction for future research.**
>
> We would like to assure you that we did not attempt to plagiarize the FTB work. We have cited it in several other places throughout our article, including:
>
> + "Notably, while our approach shares similarities with FTB in utilizing diffusion models, we differ by not requiring additional human preference datasets or task-specific score models. This reduces model bias and complexity, offering a more efficient method for inferring implicit expert preferences, making it better suited for real-world applications." (Section 5: 510–515)
> + "On the other hand, FTB (Zhang et al., 2023) utilizes an additional human preference dataset to train a task-specific score model, which calculates preferences for unlabeled trajectories. Yet, collecting extensive human preference data is labor-intensive and even challenging." (Section B.1: 770–772)
> + "Recently, some research has compared the ground-truth returns or human preferences of different trajectories to train a generative model that can improve inferior trajectories into high-quality ones, effectively enhancing the quality of the training data. (Liu & Abbeel, 2023; Zhang et al., 2023)." (Section 1: 062–065)
>
> **We will carefully review and revise to address any potential issues and ensure appropriateness, including adding a dedicated section to discuss the similarities and differences between FTB and our approach.**

---

> ### Author Response · Authors · 2024-11-15
> **Official Comment by Authors (2)**
>
> **[Writing]**
>
> **We would like to clarify that our primary goal in writing is to convey our ideas with clarity and precision.** In response to some of the concerns you raised regarding the writing, we have provided a detailed explanation of our overall writing approach.
>
> [Introduction & Background]
> These two sections mainly focus on various aspects related to the TDGBA method. Specifically, Section 3.2 presents a description of the foundational background knowledge on diffusion models, which typically shares some similarities with other papers discussing diffusion like [1]. In contrast to FTB, we provide separate discussions on offline imitation learning and Wasserstein distance in the context of other background knowledge, all of which are developed based on the TDGBA framework.
>
> [Experiment]
> In the experimental section, we compared nearly all recent offline imitation learning methods, and we made two separate chapters comparing BC-based and reward-based methods. We list the baseline methods for comparison between TDGBA and FTB here:
> ● TDGBA: (BC, %10BC, SQIL, DEMODIC, SMODICE, LobsDICE)
> ● FTB: (%10BC, TD3+BC, IQL, OPRL, PT, IPL)
>
> **It is evident that our choice of baselines is quite different.**
> In Section 4.2, we also visualize the correlation between the implicit expert preferences we defined and the ground-truth returns, **further demonstrating that we can achieve results similar to those in FTB without using human preferences**. Additionally, we plotted the distribution differences between the rewards learned by various reward-based methods and the true rewards, which, in comparison to FTB, **not only compares the true reward distribution but also clearly shows the biases in reward estimation by reward-based methods.**
>
> In the analysis of generated trajectories (Section 4.3), we compared TDGBA’s performance across various aspects using both synthetic data under unconditional preferences and original data. **We also further improved the experiment demonstrating diversity. While FTB only used a standard Gaussian distribution to show diversity, we believe this method is incomplete, as increasing Gaussian noise also increases diversity. Therefore, we switched to scatter plots combined with dynamic error under different Gaussian noise enhancements to better demonstrate this property.**
>
> [Related Work]
> In this section, we first discussed the related works on offline imitation learning. In the application of generative model in RL, **apart from the classical works such as Decision Diffuser[1], we also supplemented with works on diffusion-QL[2], diffusion policy[3], MTdiff[4] and included FTB. We believe these works are necessary to present in the paper.**
>
> [Conclution & Limitations & Future Work]
> In this part, we further clarified the unique limitations of TDGBA and explicitly stated that it may not be suitable for handling multimodal expert trajectories.
>
> In summary, in the design of the writing structure, we closely followed the presentation of TDGBA and included all the necessary related content. **We will make every effort to revise the areas you pointed out. Our primary goal is to avoid any potential concerns or misunderstandings.**
>
> **Finally, we would like to once again express our sincere gratitude to the reviewer for the time and effort invested in reviewing our work and rebuttal.  Most importantly, while we acknowledge that our work may not be fully mature or perfect and that there may be some areas with shortcomings, we want to emphasize that we have no intention of plagiarism. It would be unreasonable to label our efforts as plagiarism. We kindly ask you to reconsider our work in light of the points we have raised above. We will carefully revise and refine the work based on your valuable feedback.** We look forward to your response, as it is very important to us.
>
> Thank you.
>
>
>
> [Reference]
>
> [1] Is conditional generative modeling all you need for decision-making. ICLR, 2023.
>
> [2] Diffusion policies as an expressive policy class for offline reinforcement learning. ICLR, 2023.
>
> [3] Diffusion policy: Visuomotor policy learning via action diffusion. RSS, 2023.
>
> [4] Diffusion model is an effective planner and data synthesizer for multi-task reinforcement learning. NeurIPS, 2023.
>
> [5] Flow to better: Offline preference-based reinforcement learning via preferred trajectory generation. ICLR, 2024.

---

> ### Author Response · Authors · 2024-11-25
> **Follow-Up on Rebuttal Discussion**
>
> Dear Reviewer,
>
> We are very grateful for the time and effort you have dedicated to reviewing our work. We understand that you may be busy with your recent responsibilities, but we sincerely hope that you can provide further feedback on our rebuttal. It is of utmost importance to us to address any concerns you may have in our work.
>
> We genuinely appreciate your time and consideration and look forward to your response.
>
> Best regards,
>
> All authors

---

> ### Comment · Reviewer_oZ9p · 2024-11-25
>
> The authors argue that TDGBA and FTB are methods proposed for two different settings, and the algorithms compared in their experimental sections are entirely different. However, I still believe that the two methods share an essential similarity: both use a diffusion model to learn the process from 'bad' to 'good' trajectories, then augment the dataset with additional expert trajectories for better behavior cloning. **The only difference is TDGBA adopts a different alignment measure compared to FTB. Other than that, the details of the two algorithms are entirely identical.**
>
> Additionally, I want to clarify that I am not suggesting TDGBA lacks any value as a method. My concern is that the authors have used an algorithm that is nearly identical to FTB but have provided very limited discussion of this technical similarity in the paper (at least not in a prominent position). Moreover, the two papers also exhibit significant structural similarity, and by this, I mean the organization of the paper, not at the textual level.
>
> Finally, regardless of whether this paper is accepted at this conference or submitted to another, I strongly recommend that the authors discuss the similarities and differences between TDGBA and FTB in detail in the main text. As for whether the paper constitutes plagiarism, I still believe that the two methods are highly similar at this stage. However, the AC and other reviewers have the right to form their own opinions.

---

> > ### Author Response · Authors · 2024-11-29
> > **Official Comment by Authors (3)**
> >
> > Dear Reviewer,
> >
> > We sincerely thank you for taking the time to review our submission and for providing valuable feedback. Based on your comments, we have worked diligently within the limited time available to address the issues and have updated our submission. Below, we provide a detailed explanation of the revisions made in response to your feedback:
> >
> > **[Revise details1 — Discussion of TDGBA and FTB]**
> >
> > We added a dedicated **Discussion** section to elaborate on the similarities and differences between TDGBA and FTB. For example:
> >
> > _"TDGBA adopts methods from FTB for trajectory optimization and policy learning. However, the core contribution of our work lies in successfully combining the advantages of preference learning with offline imitation learning for the first time."_(Lines 507~519)
> >
> > **[Revise details2 — Preliminaries about FTB]**
> >
> > We have added a detailed explanation of the trajectory optimization process based on preferences and the application of diffusion models in FTB, presented in the **Preliminaries** section under the **Flow-to-Better with Trajectory Diffuser** subsection. Consequently, we have removed the corresponding content about trajectory optimization and diffusion models from the 'Method' section of the previous version.
> >
> > **[Revise details3 — Revisions to the Method section]**
> >
> > We removed Sections 2 and 3 from the original **Method** section.
> >
> > Section 2 was replaced with **'Implicit expert preferences'**, which discusses the similarities between implicit expert preferences and FTB in estimating preference values, as well as their correlation with ground-truth returns. We also highlighted potential challenges in FTB related to preference prediction accuracy under varying numbers of preferred trajectories. Additionally, we emphasized the simplicity and practicality of the proposed metric.
> >
> > Section 3, focusing on the **Algorithm**, has been revised with more detailed annotations to enhance clarity.
> >
> > **[Revise details4 — Emphasis and citations of FTB]**
> >
> > We have emphasized and cited FTB in multiple sections, including the **Introduction** and **Related Work**. Examples include:
> >
> > + _"Recently, Flow-to-Better (FTB) introduced a method xxx"_ (Lines 063–068)
> > + _"We adopt the trajectory optimization approach proposed in FTB, xxx"_ (Lines 073–074)
> > + _"By integrating these with the trajectory optimization approach in FTB, xxx"_ (Lines 115–116)
> >
> > **[Revise details5 — Structural revisions]**
> >
> > We have reorganized the overall structure of the paper to improve its coherence and readability, for example:
> >
> > + Removed sections on diffusion and generative models from the original **'Background'** and **'Related Work'** chapters.
> > + Adjusted the layout of the **'Experiment'** section. While some minor parts were removed, we believe the overall structure and content are necessary. Additionally, the structure remains consistent with related works, including those beyond FTB.
> >
> > **[Discussion of our contribution]**
> >
> > > The only difference is that TDGBA adopts a different alignment measure compared to FTB. Other than that, the details of the two algorithms are entirely identical."
> >
> > We respectfully disagree with this perspective. While we leveraged FTB’s trajectory optimization process as it represents the state-of-the-art in offline preference RL, our main contribution lies in introducing the concept of **implicit expert preferences**. For the first time, we have applied the advantages of preference learning combined with trajectory optimization to the offline imitation learning (IL) domain. Furthermore, we believe our proposed preference framework is theoretically compatible with other preference learning methods. Due to time constraints, we have not yet completed experiments to verify this, but we plan to do so in future revised work.
> >
> > Finally, we kindly hope that you revise your previous review and remove any references to phrases like _"this paper is suspected of plagiarism"_ or _"this paper has plagiarized another paper"_. Given the public nature of ICLR, such statements could have a severe negative impact on our work and future research. We assure you that we have made every effort to address the concerns raised. Regardless of the final outcome, we hope you can reconsider your assessment and revise your previous comments to reflect the current state of the paper.
> >
> > Thank you again for your understanding and constructive suggestions. We truly appreciate your feedback and look forward to your response.
> >
> > Best regards,
> > All authors

---

> ### Author Response · Authors · 2024-12-03
> **Official Comment by Authors (4)**
>
> Dear Reviewer:
>
> We would like to express our sincere gratitude for the effort you have put into reviewing our work. We fully understand that, due to your busy schedule, you may not have had the time to respond. As the discussion phase is nearing its end, we sincerely hope that you can review our updated work and revise your previous comments accordingly, as this would be of great importance to us.
>
> Once again, we sincerely appreciate your valuable time and look forward to your response.
>
> Best regards,
>
> All authors

---

### Meta-Review · Area_Chair_2hQz · 2024-12-20

**Metareview:**

(a) Summary: This paper proposes a method called Trajectory-level Data Generation with Better Alignment (TDGBA) for offline imitation learning (IL).
(b) Strengths: The paper is generally well-written. The proposed method achieves better performance than baselines.
(c) Weaknesses: The reviewers pointed out multiple major concerns and issues of this paper, such as lack of novelty, missing explanations of technical details and experimental results.
(d) The majority of reviewers pointed out major issues of this paper, which were not fully addressed by the authors. Major improvement is required.

**Additional Comments On Reviewer Discussion:**

Although the authors provided more details and explanations during the rebuttal, there remain a few major concerns that were not fully resolved after the discussion.

---

### Decision · Program_Chairs · 2025-01-22

Reject